# Adaptive Robust Vehicle Motion Control for Future Over-Actuated Vehicles [†]

**Moad Kissai [1],[\*]** , **Bruno Monsuez [1]**, **Xavier Mouton [2]**, **Didier Martinez [2]** and **Adriana Tapus [1]**

[1] Autonomous Systems and Robotics Lab, Computer Science and System Engineering Department (U2IS), École Nationale Supérieure de Techniques Avancées (ENSTA ParisTech), Institut Polytechnique de Paris (IP Paris), 828 Boulevard des Maréchaux, 91120 Palaiseau, France; bruno.monsuez@ensta-paristech.fr (B.M.); adriana.tapus@ensta-paristech.fr (A.T.)

[2] Chassis Systems Department, Groupe Renault, 1 Avenue du Golf, 78280 Guyancourt, France; xavier.mouton@renault.com (X.M.); didier.d.martinez@renault.com (D.M.)

[\*] Correspondence: moad.kissai@ensta-paristech.fr; Tel.: +33-6-62153444

[†] This paper is an extended version of our paper published in Kissai, M.; Monsuez, B.; Tapus, A.; Mouton, X.; Martinez, D. Gain-Scheduled $\mathcal{H}_\infty$ for Vehicle High-Level Motion Control. In Proceedings of the 6th International Conference on Control, Mechatronics and Automation (ICCMA 2018), Tokyo, Japan, 12–14 October 2018; pp. 97–104.

**Abstract:** Many challenges still need to be overcome in the context of autonomous vehicles. These vehicles would be over-actuated and are expected to perform coupled maneuvers. In this paper, we first discuss the development of a global coupled vehicle model, and then we outline the control strategy that we believe should be applied in the context of over-actuated vehicles. A gain-scheduled $\mathcal{H}_\infty$ controller and an optimization-based Control Allocation algorithms are proposed. High-fidelity co-simulation results show the efficiency of the proposed control logic and the new possibilities that could offer. We expect that both car manufacturers and equipment suppliers would join forces to develop and standardize the proposed control architecture for future passenger cars.

**Keywords:** $\mathcal{H}_\infty$ control; gain scheduling; control allocation; vehicle dynamics; robustness; co-simulation

---

## 1. Introduction

The automotive sector is on the move. Each day, we get closer and closer to the revolution of autonomous vehicles, which is one of the most complex systems. This is a topic still with many challenges to overcome. Decision-making algorithms still need to cover more critical scenarios, sensors still need to become more reliable, control strategies still need to improve their robustness and adaptability and so on. In the field of ground vehicles, the only effector is the tire. Its potential varies with friction and vertical load [1]. Moreover, longitudinal and lateral forces are coupled, and their behavior is highly nonlinear. To top it all off, couplings exist also at the vehicle level. Roll dynamics can influence yaw rate dynamics, and lateral speed can penalize the longitudinal one, etc. [2]. In addition, the dynamics at low speed differ from the ones at high speed. To prove the safety of autonomous vehicles, the control of dynamics at the limit of the vehicle's handling should be ensured [3]. Another problem concerns the vehicle's parameter uncertainties. The overall mass can be uncertain and may vary [4], the tires' cornering stiffness can be uncertain and can also age [5], and so on. The control strategy should stay valid whether there is one passenger or many in the vehicle, whether the road is dry or wet, whether tires are new or start to wear, etc. An adaptive robust controller is needed, especially when different embedded systems are involved making the vehicle over-actuated.

In this context, in [6], a Relative Gain Array (RGA) study has been conducted to evaluate the system couplings near the crossover frequency using a simplified four-wheeled vehicle equipped

with an Electronic Stability Program (ESP) and an Electric Power Assisted Steering (EPAS). Authors concluded that the system can be decoupled for high frequencies. Youla's parameterization has been used then for each Single-Input Single-Output (SISO) transfer function. Results were acceptable for each variable only when the throttle was on, i.e., with the presence of a driver to control the longitudinal speed. Authors in [7] proposed a Sliding Mode Control (SMC) to coordinate the ESP and active steering devices, both in front and rear. The bicycle model has been chosen to design the controller. Two objectives have been pursued: maneuverability by means of yaw rate tracking, and lateral stability by minimizing the side-slip angle. A four-wheeled vehicle model has been considered afterwards in the low-level control for the ESP. Couplings were not managed at the high-level control as two different vehicle models have been considered. Moreover, no lateral velocity control has been ensured whereas a vehicle equipped with Active Front and Rear Steering (AFS & ARS) can ensure a lateral transitional motion to avoid an obstacle, for example. SMC was used also in [8] to control an electric vehicle equipped with a four-Wheel Steering (4WS) system and a four-Wheel Drive (4WD) system. A four-wheeled vehicle model was considered, but only to control the lateral dynamics of the vehicle. Couplings with the longitudinal dynamics were then ignored. In [9], a Linear with Parameters Varying (LPV)/$\mathcal{H}_\infty$ controller has been chosen as the high-level controller for a vehicle equipped with AFS and rear braking. Although good robustness is ensured, again, only a bicycle model was considered to design the controller. A 14-Degrees Of Freedom (DOF) full vehicle model equipped with an AFS, an Anti-lock Braking System (ABS), and Semi-Active Suspension (SAS) has been used in [2] and then simplified for control synthesis. A high-level controller based on SMC has been chosen. As the authors noted, the SMC procedure suffers from high-frequency chattering. The sign function can be used instead of the saturation function [2] to reduce the effect of chattering. We believe that this method is more suitable for electronic devices, and in contrast, could accelerate mechanical actuators aging or tire wear.

The state-of-the-art shows that either we employ complex robust controllers based on simplified non-coupled vehicle models, or we decouple a complex vehicle model to use simplified controllers at each direction. In [10], the authors prioritise neither the first nor the second approach. A new approach is rather investigated where a relatively complex robust high-level controller is used based on a relatively complex four-wheeled vehicle model with an optimal coordination strategy. The goal is to evaluate the dynamic couplings at the vehicle level to justify the structure of the high-level controller. A bicycle model depicting the lateral dynamics of the vehicle can only serve to develop a lateral control strategy. For a coupled longitudinal–lateral control, a coupled longitudinal–lateral vehicle model is needed. This journal paper is actually an extended version of [10]. A major difference here is the presentation of a new global vehicle dynamics model that can serve the readers in various control problems. The model is then simplified to fit our specific model. The control strategy is then explained and detailed so the reader can reproduce and adapt the exposed design steps to this problem.

The rest of this paper is structured as follows: We start in Section 2 by developing a global vehicle model that can help control engineers study the dynamic couplings of the vehicle. In Section 3, the vehicle motion control strategy is synthesized using a highly over-actuated vehicle. Section 4 presents results obtained by co-simulation of Matlab/Simulink$^\circledR$ and AMESim$^\circledR$. A discussion about open challenges in terms of robustness and qualitative objectives is provided in Section 5. Conclusions and future works are outlined in Section 6.

## 2. Vehicle Modelling

Future Autonomous Vehicles (AV) have to handle longitudinal, lateral, and eventually, vertical control at the same time. These dynamics are coupled since they are related to the same system. For a Global Chassis Control (GCC), the vehicle's internal dynamic couplings should be taken into account. These couplings could be more restrictive, or in contrast, more relaxed depending on the chassis systems integrated within the same vehicle. For example, a two-Wheel Steering (2WS) vehicle can only avoid an obstacle by changing its heading (yaw angle). Unlike a 2WS vehicle, a 4WS vehicle, by steering

the four wheels in the same direction, can avoid an obstacle without changing its heading. We still do not know the future hardware design of passenger cars, and because the design can differ from a car manufacturer to another, here, we develop a new detailed global vehicle modelling. This vehicle model can be reduced afterwards depending on the chassis systems integrated and the possible states to be controlled, as it is done in [10]. For a proper construction of vehicle motion equations, we adopt the ISO 8855-2011 (https://www.iso.org/standard/51180.html) shown in Figure 1.

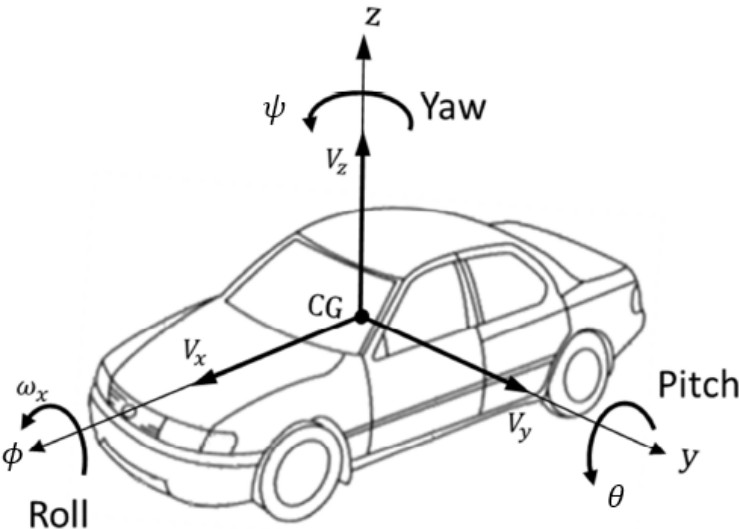

**Figure 1.** Vehicle Axis System ISO 8855-2011.

### 2.1. Multi-Body Approach

In order to take into account the dynamic couplings, vertical load transfer, influence of suspensions and so on, the vehicle will be broken down into two supposedly undeformable masses: the sprung mass (Includes the vehicle body, engine, passengers and so on.), and the unsprung mass (Includes the wheels, suspensions, brakes and so on.), as Figure 2 shows.

In addition, to take into account the differences between the influence of the front axle and the rear axle, the unsprung mass is also decomposed into two supposedly undeformable masses. We then have $\Sigma = S_s + S_{uf} + S_{ur}$, with:

- $\Sigma$ : the overall vehicle of a mass $M$ and a Center of Gravity (CoG) $G$,
- $S_s$ : the sprung mass of a mass $M_s$ and a CoG $G_s$,
- $S_{uf}$: the front unsprung mass of a mass $M_{uf}$ and a CoG $G_{uf}$,
- $S_{ur}$ : the rear unsprung mass of a mass $M_{ur}$ and a CoG $G_{ur}$.

Therefore, by noting "$\left\{ \mathcal{D}\left( \Sigma/\mathcal{R}_g \right) \right\}_G$" the dynamic torsor of the vehicle with respect to the inertial frame of reference $\mathcal{R}_g$ at the point $G$, we get:

$$\left\{ \mathcal{D}\left( \Sigma/\mathcal{R}_g \right) \right\}_G = \left\{ \mathcal{D}\left( S_s/\mathcal{R}_g \right) \right\}_G + \left\{ \mathcal{D}\left( S_{uf}/\mathcal{R}_g \right) \right\}_G + \left\{ \mathcal{D}\left( S_{ur}/\mathcal{R}_g \right) \right\}_G \tag{1}$$

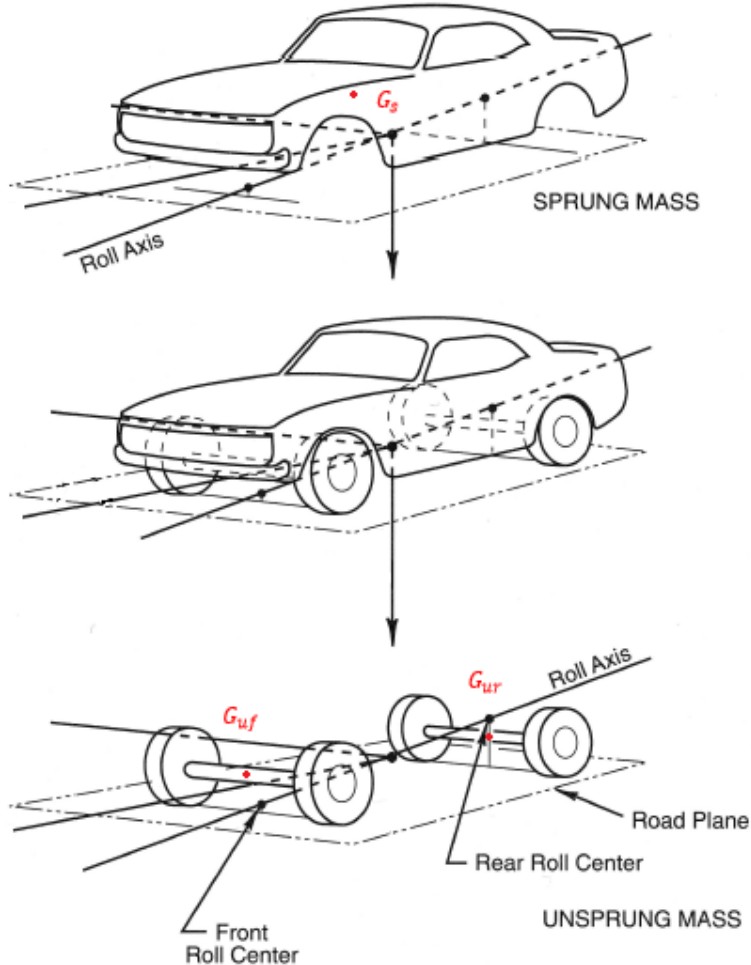

**Figure 2.** The sprung and unsprung masses decomposition.

### 2.2. Linear Equations of Motion

Since the CoG of the sprung mass can move with respect to the unsprung mass, it is simpler to start establishing the equations of motion with respect to a fixed point rather than the moving CoG [11]. Here, we start the calculation by considering the roll center that we note O. The velocity of this point is noted:

$$\vec{V}\left(O/\mathcal{R}_g\right) = V_{O_x}\vec{i} + V_{O_y}\vec{j} \tag{2}$$

With $\vec{i}$ and $\vec{j}$ as the unit vectors of the chassis frame in the longitudinal and lateral direction, respectively. Therefore, $V_{O_x}$ and $V_{O_y}$ are the longitudinal and lateral velocity of the point $O$ respectively.

The chassis frame moves with respect to the inertial frame, and the unsprung mass is not subject to neither the roll nor the pitch motions. The acceleration of the roll center is then:

$$
\vec{\Gamma}\left(O/\mathcal{R}_g\right) = \begin{bmatrix} \dot{V}_{O_x} \\ \dot{V}_{O_y} \\ 0 \end{bmatrix} + \begin{bmatrix} 0 \\ 0 \\ \dot{\psi} \end{bmatrix} \wedge \begin{bmatrix} V_{O_x} \\ V_{O_y} \\ 0 \end{bmatrix}
$$

$$
= \left(\dot{V}_{O_x} - \dot{\psi}V_{O_y}\right)\vec{i} + \left(\dot{V}_{O_y} + \dot{\psi}V_{O_x}\right)\vec{j} \tag{3}
$$

where $\dot{\psi}$ is the vehicle's yaw rate, and $\wedge$ denotes the cross product.

The same procedure can be applied to the points $G_{uf}$ and $G_{ur}$. For the front unsprung mass:

$$\begin{cases} \vec{V}\left(G_{uf}/\mathcal{R}_g\right) = V_{O_x}\vec{i} + \left(\dot{V}_{O_y} + \dot{\psi}l_f\right)\vec{j} \\ \vec{\Gamma}\left(G_{uf}/\mathcal{R}_g\right) = \left(\dot{V}_{O_x} - \dot{\psi}V_{O_y} - l_f\dot{\psi}^2\right)\vec{i} + \left(\dot{V}_{O_y} + \dot{\psi}V_{O_x} + l_f\ddot{\psi}\right)\vec{j} \end{cases} \tag{4}$$

With $l_f$ is the horizontal distance between $G$ and $G_{uf}$. For the rear unsprung mass:

$$\begin{cases} \vec{V}\left(G_{ur}/\mathcal{R}_g\right) = V_{O_x}\vec{i} + \left(V_{O_y} - \dot{\psi}l_r\right)\vec{j} \\ \vec{\Gamma}\left(G_{ur}/\mathcal{R}_g\right) = \left(\dot{V}_{O_x} - \dot{\psi}V_{O_y} + l_r\dot{\psi}^2\right)\vec{i} + \left(\dot{V}_{O_y} + \dot{\psi}V_{O_x} - l_r\ddot{\psi}\right)\vec{j} \end{cases} \tag{5}$$

With $l_r$ is the horizontal distance between $G$ and $G_{ur}$.

Regarding the sprung mass, the calculation is slightly more complicated as the vehicle's body turns with respect to the unsprung mass. The roll angle $\phi$ and the pitch angle $\theta$ appear. The relationship between the vehicle's body frame and the chassis frame is:

$$\begin{pmatrix} \vec{i}_s \\ \vec{j}_s \\ \vec{k}_s \end{pmatrix} = \begin{pmatrix} \cos\theta & 0 & -\sin\theta \\ \sin\phi\sin\theta & \cos\phi & \sin\phi\cos\theta \\ \cos\phi\sin\theta & -\sin\phi & \cos\phi\cos\theta \end{pmatrix} \begin{pmatrix} \vec{i} \\ \vec{j} \\ \vec{k} \end{pmatrix} \tag{6}$$

We then obtain:

$$\vec{V}\left(G_s/\mathcal{R}_g\right) = \begin{bmatrix} V_{O_x} - \dot{\theta}\left(l_s\sin\theta + h_s\cos\phi\cos\theta\right) + \dot{\phi}h_s\sin\phi\sin\theta - \dot{\psi}h_s\sin\phi \\ V_{O_y} + \dot{\phi}h_s\cos\phi + \dot{\psi}\left(l_s\cos\theta - h_s\cos\phi\sin\theta\right) \\ \dot{\theta}\left(l_s\cos\theta - h_s\cos\phi\sin\theta\right) - \dot{\phi}h_s\sin\phi\cos\theta \end{bmatrix} \tag{7}$$

With $l_s$ and $h_s$ are the horizontal and vertical distances between $G_s$ and $O$, respectively.

However, as expected, due to the transformation in Equation (6), the acceleration equations obtained are very large to expose. Nevertheless, we can propose at this point several simplifications. We assume relatively small values of the roll and pitch angles and angular velocities with respect to yaw dynamics. This gives:

$$\begin{cases} \sin\phi \approx \phi & \cos\phi \approx 1 & \sin\theta \approx \theta & \cos\theta \approx 1 \\ \phi\theta \approx 0 & \dot{\phi}\dot{\theta} \approx 0 & \dot{\phi}^2 \approx 0 & \dot{\theta}^2 \approx 0 \end{cases} \tag{8}$$

In addition, to be able to apply the dynamic resultant theorem [12] (Newton's second law of motion):

$$M\vec{\Gamma}\left(G/\mathcal{R}_g\right) = \vec{F}\left(\bar{\Sigma} \to \Sigma\right) \tag{9}$$

with $\vec{F}$ representing the exterior forces applied to $\Sigma$ and $\bar{\Sigma}$ the complement of the system $\Sigma$, we have to bring the calculation to a single point: $G$. To do so, we make use of the definition of the center of mass:

$$\overrightarrow{OG} = \frac{\sum_i m_i \overrightarrow{OG_i}}{\sum_i m_i} \tag{10}$$

Therefore, with the simplifications in (8):

$$M\vec{\Gamma}\left(G/\mathcal{R}_g\right) = M_s\vec{\Gamma}\left(G_s/\mathcal{R}_g\right) + M_{uf}\vec{\Gamma}\left(G_{uf}/\mathcal{R}_g\right) + M_{ur}\vec{\Gamma}\left(G_{ur}/\mathcal{R}_g\right)$$

$$= \begin{bmatrix} M\left(\dot{V}_{O_x} - \dot{\psi}V_{O_y}\right) - \ddot{\theta}M_s\left(l_s\theta + h_s\right) - \ddot{\psi}M_sh_s\phi + \dot{\psi}^2Mh_g\theta - 2\dot{\phi}\dot{\psi}M_sh_s \\ M\left(\dot{V}_{O_y} + \dot{\psi}V_{O_x}\right) + \ddot{\phi}M_sh_s - \ddot{\psi}Mh_g\theta - \dot{\psi}^2M_sh_s\phi - 2\dot{\theta}\dot{\psi}M_s\left(l_s\theta + h_s\right) \\ \ddot{\theta}M_s\left(l_s - h_s\theta\right) - \ddot{\phi}M_sh_s\phi \end{bmatrix} \tag{11}$$

With $h_g$ is the horizontal distance between O and G.

For the exterior forces, let us consider Figure 3.

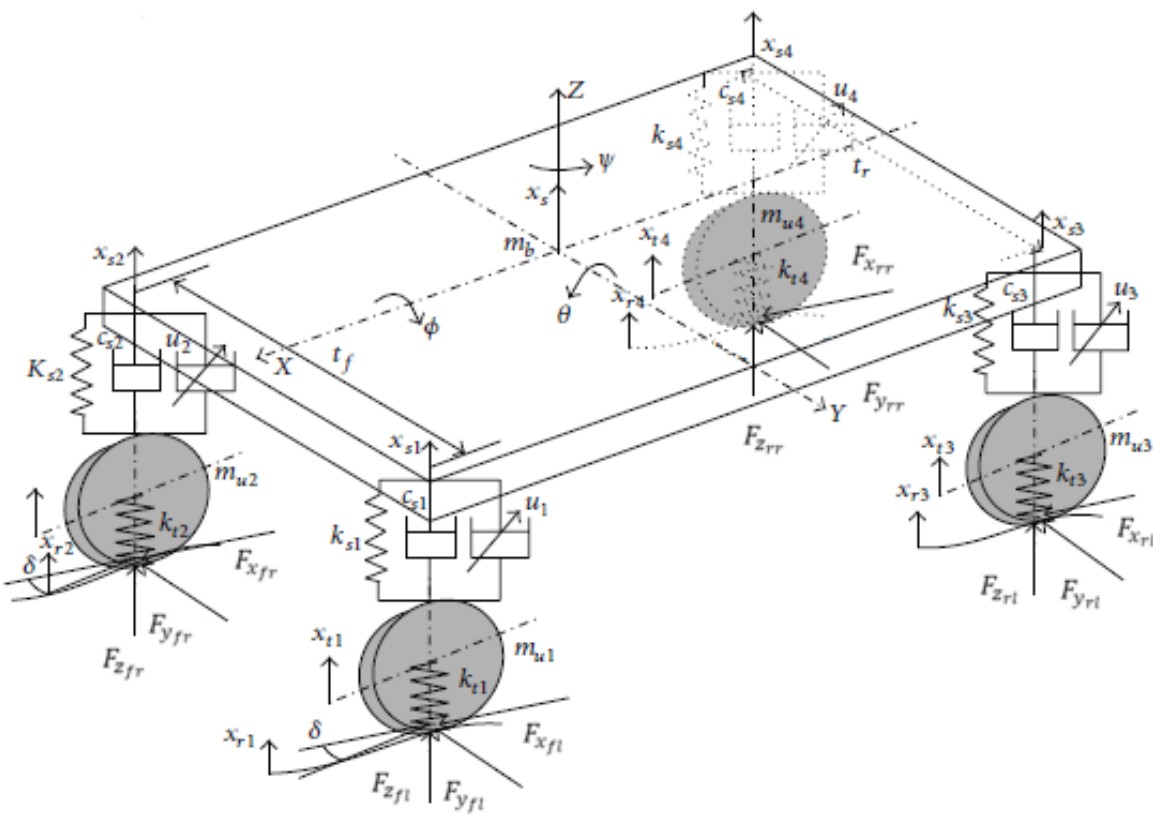

**Figure 3.** 1four-DoF vehicle dynamic model (adapted from [2]).

If we consider the overall system $\Sigma$, the exterior forces are:

- $F_{x_{i,j}}$ : $i - j$ longitudinal tire force (Where "$i$" is front or rear, and "$j$" is right or left.),
- $F_{y_{i,j}}$ : $i - j$ lateral tire force,
- $F_{z_{i,j}}$ : vertical load on the $i - j$ tire,
- $\vec{P}$ : the vehicle's weight.

Notice that we do not take into account the aerodynamic forces for example. These forces are not controllable and would be considered as disturbances. They should be rejected by the robust high-level controller. We then apply the Equation (9) using the formulas obtained in Equation (11) to get the linear equations of motion:

$$M \left( \dot{V}_{O_x} - \dot{\psi} V_{O_y} \right) - \ddot{\theta} M_s \left( l_s \theta + h_s \right) - \ddot{\psi} M_s h_s \phi + \dot{\psi}^2 M h_g \theta - 2\dot{\phi}\dot{\psi} M_s h_s$$
$$= \left( F_{x_{f,l}} + F_{x_{f,r}} \right) \cos \delta_f + \left( F_{x_{r,l}} + F_{x_{r,r}} \right) \cos \delta_r \tag{12}$$
$$- \left( F_{y_{f,l}} + F_{y_{f,r}} \right) \sin \delta_f - \left( F_{y_{r,l}} + F_{y_{r,r}} \right) \sin \delta_r$$

$$M \left( \dot{V}_{O_y} + \dot{\psi} V_{O_x} \right) + \ddot{\phi} M_s h_s - \ddot{\psi} M h_g \theta - \dot{\psi}^2 M_s h_s \phi - 2\dot{\theta}\dot{\psi} M_s \left( l_s \theta + h_s \right)$$
$$= \left( F_{y_{f,l}} + F_{y_{f,r}} \right) \cos \delta_f + \left( F_{y_{r,l}} + F_{y_{r,r}} \right) \cos \delta_r \tag{13}$$
$$+ \left( F_{x_{f,l}} + F_{x_{f,r}} \right) \sin \delta_f + \left( F_{x_{r,l}} + F_{x_{r,r}} \right) \sin \delta_r$$

$$\ddot{\theta} M_s \left( l_s - h_s \theta \right) - \ddot{\phi} M_s h_s \phi = Mg - F_{z_{f,l}} - F_{z_{f,r}} - F_{z_{r,l}} - F_{z_{r,r}} \tag{14}$$

where $\delta_f$ and $\delta_r$ are the front and rear steering angles respectively.

In order to show the influence of the suspensions, we should isolate only the sprung mass where the suspension forces are at the exterior of the studied system. In case of active suspensions, as it is the case in Figure 3, we have [2]:

$$\begin{cases} F_{s_{f,l}} = k_{s_{f,l}} \left( z_{p_{f,l}} - z_{s_{f,l}} \right) + c_{s_{f,l}} \left( \dot{z}_{p_{f,l}} - \dot{z}_{s_{f,l}} \right) - \dfrac{k_{\phi_f}}{2t_f} \left( \phi - \dfrac{z_{p_{f,l}} - z_{s_{f,l}}}{2t_f} \right) + u_{f,l} & (15) \\[2.5ex] F_{s_{f,r}} = k_{s_{f,r}} \left( z_{p_{f,r}} - z_{s_{f,r}} \right) + c_{s_{f,r}} \left( \dot{z}_{p_{f,r}} - \dot{z}_{s_{f,r}} \right) + \dfrac{k_{\phi_f}}{2t_f} \left( \phi - \dfrac{z_{p_{f,r}} - z_{s_{f,r}}}{2t_f} \right) + u_{f,r} & (16) \\[2.5ex] F_{s_{r,l}} = k_{s_{r,l}} \left( z_{p_{r,l}} - z_{s_{r,l}} \right) + c_{s_{r,l}} \left( \dot{z}_{p_{r,l}} - \dot{z}_{s_{r,l}} \right) + \dfrac{k_{\phi_r}}{2t_r} \left( \phi - \dfrac{z_{p_{r,l}} - z_{s_{r,l}}}{2t_r} \right) + u_{r,l} & (17) \\[2.5ex] F_{s_{r,r}} = k_{s_{r,r}} \left( z_{p_{r,r}} - z_{s_{r,r}} \right) + c_{s_{r,r}} \left( \dot{z}_{p_{r,r}} - \dot{z}_{s_{r,r}} \right) - \dfrac{k_{\phi_r}}{2t_r} \left( \phi - \dfrac{z_{p_{r,r}} - z_{s_{r,r}}}{2t_r} \right) + u_{r,r} & (18) \end{cases}$$

where:

- $z_{p_i}$ : vertical travel of tires,
- $z_{s_i}$ : vertical travel of suspensions,
- $k_{s_i}$ : suspension's stiffness,
- $c_{s_i}$ : suspension's damping,
- $k_{\phi_f}, k_{\phi_r}$ : the front and rear anti-roll bars stiffness respectively,
- $t_f, t_r$ : the front and rear track of the vehicle respectively,
- $u_{s_i}$ : control forces of the active suspensions.

Using again the theorem (9), we can get:

$$M_s \left[ \ddot{\theta} \left( l_s - h_s \theta \right) - \ddot{\phi} h_s \phi \right] = M_s g - F_{s_{f,l}} - F_{s_{f,r}} - F_{s_{r,l}} - F_{s_{r,r}} \tag{19}$$

## 2.3. Angular Equations of Motion

First, we calculate the angular moment with respect to the reference point $O$ and for each undeformable mass apart. The definition of the angular moment applied to the front unsprung mass is as follows [12]:

$$\vec{\sigma} \left( O, S_{uf}/\mathcal{R}_g \right) = M_{uf} \overrightarrow{OG_{uf}} \wedge \vec{V} \left( O/\mathcal{R}_g \right) + \overline{\overline{I_{uf}}} \left( O, S_{uf} \right) \cdot \vec{\Omega}_c \tag{20}$$

where $I_{uf}$ is the inertia tonsor of the mass $S_{uf}$. Its definition applied to any vector $\vec{u}$ at the point $O$ is:

$$\overline{\overline{I_{uf}}} \left( O, S_{uf} \right) . \vec{u} = - \int_{P \in S_{uf}} \left[ \overrightarrow{OP} \wedge \left( \overrightarrow{OP} \wedge \vec{u} \right) \right] dm \tag{21}$$

The vector $\vec{\Omega}_c$ is the angular velocity vector, which in this case contains only the yaw rate.

The dynamic moment $\vec{\delta}$ defined in any point $A$, is deduced from the angular moment using the following definition [12]:

$$\vec{\delta} \left( A, S/\mathcal{R}_g \right) = \left. \dfrac{d\vec{\sigma} \left( A, S/\mathcal{R}_g \right)}{dt} \right|_{\mathcal{R}_g} + M\vec{V} \left( A/\mathcal{R}_g \right) \wedge \vec{V} \left( G/\mathcal{R}_g \right) \tag{22}$$

By choosing $A = G$, this definition is simplified into:

$$\vec{\delta} \left( G, \Sigma/\mathcal{R}_g \right) = \left. \dfrac{d\vec{\sigma} \left( G, \Sigma/\mathcal{R}_g \right)}{dt} \right|_{\mathcal{R}_g} \tag{23}$$

We use after the theorem of Huygens–Steiner defined as follows [12]: In a solid with a mass $m$, if two axis are parallel with a distance $d$ between them (see Figure 4), with one of the axis $(D)$ passes through the CoG $G$, and the second one is noted $(\Delta)$, then we have:

$$\overline{I_{/\Delta}}.\vec{u} = \overline{I_{/D}}.\vec{u} + md^2 \tag{24}$$

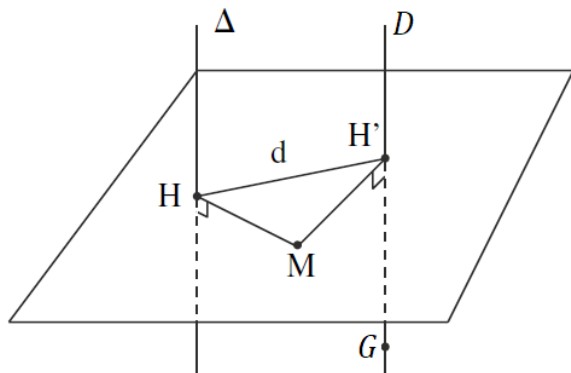

**Figure 4.** Huygens's theorem (adapted from [12]).

Using this theorem to $\overline{I_{uf}}$ we obtain:

$$\overline{I_{uf}}\left(O, S_{uf}\right) = \begin{pmatrix} I_{x_{uf}} + M_{uf}h_{uf}^2 & 0 & -I_{xz_{uf}} + M_{uf}l_f h_{uf} \\ 0 & I_{y_{uf}} + M_{uf}\left(l_f^2 + h_{uf}^2\right) & 0 \\ -I_{xz_{uf}} + M_{uf}l_f h_{uf} & 0 & I_{z_{uf}} + M_{uf}l_f^2 \end{pmatrix} \tag{25}$$

With $I_{x_{uf}}$, $I_{y_{uf}}$, and $I_{z_{uf}}$ are the inertia moment in the longitudinal, lateral, and vertical direction with respect to the point $G_{uf}$ respectively. The zeros are due to the fact that $S_{uf}$ is symmetric with respect to the plan $\left(G_{uf}, x, z\right)$. The additional terms are due to the theorem of Huygens–Steiner and the fact that the expressions have been brought to the point $O$. We finally get:

$$\vec{\sigma}\left(O, S_{uf}/\mathcal{R}_g\right) = \begin{bmatrix} M_{uf}h_{uf}\left(V_{Oy} + l_f\dot{\psi}\right) - I_{xzuf}\dot{\psi} \\ -M_{uf}h_{uf}V_{Ox} \\ M_{uf}l_f\left(V_{Oy} + l_f\dot{\psi}\right) + I_{zuf}\dot{\psi} \end{bmatrix} \tag{26}$$

Using the same procedure we can find:

$$\vec{\sigma}\left(O, S_{ur}/\mathcal{R}_g\right) = \begin{bmatrix} M_{ur}h_{ur}\left(V_{Oy} - l_r\dot{\psi}\right) - I_{xzur}\dot{\psi} \\ -M_{ur}h_{ur}V_{Ox} \\ -M_{ur}l_r\left(V_{Oy} - l_r\dot{\psi}\right) + I_{zur}\dot{\psi} \end{bmatrix} \tag{27}$$

For the sprung mass, the equations are more complicated because of the relationship (6) and because the angular velocity vector is more sophisticated:

$$\vec{\Omega}\left(S_s/\mathcal{R}_c\right) = \dot{\phi}\vec{i}_s + \dot{\theta}\vec{j}_s + \dot{\psi}\vec{k} \tag{28}$$

where $\mathcal{R}_c$ is the vehicle's body frame. In a lateral acceleration, the pitch axis is inclined. The same remark stands for the roll axis in case of a longitudinal acceleration. The yaw axis remains the same. We then have:

$$\vec{\Omega}\left(S_s/\mathcal{R}_g\right) = \begin{bmatrix} \dot{\phi}\cos\theta + \dot{\theta}\sin\phi\sin\theta \\ \dot{\theta}\cos\phi \\ -\dot{\phi}\sin\theta + \dot{\theta}\sin\phi\cos\theta + \dot{\psi} \end{bmatrix} \tag{29}$$

The expression of the angular moment in this case is too long, and its derivative (to obtain the dynamic angular moment) is even more. The same simplifications as in (8) can be applied to moderate the results.

In addition, we should again bring the expressions to the point $G$:

$$\vec{\sigma}\left(G, S_s/\mathcal{R}_g\right) = \vec{\sigma}\left(O, S_s/\mathcal{R}_g\right) + M_s\vec{V}\left(O/\mathcal{R}_g\right) \wedge \overrightarrow{OG} \tag{30}$$

$$\vec{\sigma}\left(G, S_{uf}/\mathcal{R}_g\right) = \vec{\sigma}\left(O, S_{uf}/\mathcal{R}_g\right) + M_{uf}\vec{V}\left(O/\mathcal{R}_g\right) \wedge \overrightarrow{OG} \tag{31}$$

$$\vec{\sigma}\left(G, S_{ur}/\mathcal{R}_g\right) = \vec{\sigma}\left(O, S_{ur}/\mathcal{R}_g\right) + M_{ur}\vec{V}\left(O/\mathcal{R}_g\right) \wedge \overrightarrow{OG} \tag{32}$$

And using again the torsor properties:

$$\vec{\sigma}\left(G, \Sigma/\mathcal{R}_g\right) = \vec{\sigma}\left(G, S_s/\mathcal{R}_g\right) + \vec{\sigma}\left(G, S_{uf}/\mathcal{R}_g\right) + \vec{\sigma}\left(G, S_{ur}/\mathcal{R}_g\right) \tag{33}$$

Noting $I_{i_k}$ the inertia moment in the direction $i$ of the mass $S_k$ with respect to its CoG, and $I_{ij_k}$ the inertia moment in the plan $ij$ of the mass $S_k$ with respect to its CoG, we obtain the dynamic moment at the point $G$:

$$\delta_{G_x} = \ddot{\phi}\left(I_{x_s} + I_{xz_s}\theta + M_s h_s^2\right) + \dot{\phi}\dot{\psi}M_s h_s\phi\left(2h_s\theta - l_s\right) + \ddot{\theta}\phi\left[I_{x_s}\theta - I_{xz_s} + M_s l_s\left(l_s\theta + h_s\right)\right] + \dot{\theta}\dot{\psi}M_s\left[\left(l_s^2 - h_s^2\right) - 4l_s h_s\theta\right]$$
$$- \ddot{\psi}\left\{I_{xz_s} + I_{xz_{uf}} + I_{xz_{ur}} - M_{uf}h_{uf}l_f + M_{ur}h_{ur}l_r - M_s\left[\left(l_s 2 - h_s^2\right)\theta + l_s h_s\right]\right\} \tag{34}$$

$$\delta_{G_y} = \ddot{\theta}\left(I_{y_s} + M_s l_s^2\right) \tag{35}$$

$$\delta_{G_z} = -\ddot{\phi}\left[I_{z_s}\theta + I_{xz_s} - M_s h_s\left(l_s - h_s\theta\right)\right] + \ddot{\theta}\phi\left[I_{z_s} - I_{xz_s}\theta + M_s\left(l_s^2 - l_s h_s\theta\right)\right]$$
$$+ 2\dot{\phi}\dot{\psi}M_s h_s\phi\left(l_s\theta + h_s\right) - \dot{\theta}\dot{\psi}M_s\left[2\theta\left(l_s^2 - h_s^2\right) + 2l_s h_s\right]$$
$$+ \ddot{\psi}\left[I_{z_s} + I_{z_{uf}} + I_{z_{ur}} + M_{uf}l_f^2 + M_{ur}l_r^2 + M_s\left(l_s - h_s\theta\right)^2\right] \tag{36}$$

We then apply the dynamic moment theorem [12] (Newton's second law of motion applied to a system in rotation):

$$\vec{\delta}\left(G, \Sigma/\mathcal{R}_g\right) = \vec{M}\left(G, \overline{\Sigma} \to \Sigma\right) \tag{37}$$

With $\vec{M}$ being the exterior moments applied to the system $\Sigma$. We then obtain:

$$\delta_{G_x} = Mgh_g\phi + \frac{t_f}{2}\left(F_{z_{f,l}} - F_{z_{f,r}}\right) + \frac{t_r}{2}\left(F_{z_{r,l}} - F_{z_{r,r}}\right) - \left(h_O + h_g\right)\left[\left(F_{y_{f,l}} + F_{y_{f,r}}\right)\cos\delta_f + \left(F_{y_{r,l}} + F_{y_{r,r}}\right)\cos\delta_r\right]$$
$$- \left(h_O + h_g\right)\left[\left(F_{x_{f,l}} + F_{x_{f,r}}\right)\sin\delta_f + \left(F_{x_{r,l}} + F_{x_{r,r}}\right)\sin\delta_r\right] \tag{38}$$

$$\delta_{G_y} = Mgh_g\theta + l_f\left(F_{z_{f,l}} + F_{z_{f,r}}\right) - l_r\left(F_{z_{r,l}} + F_{z_{r,r}}\right) + \left(h_O + h_g\right)\left[\left(F_{x_{f,l}} + F_{x_{f,r}}\right)\cos\delta_f + \left(F_{x_{r,l}} + F_{x_{r,r}}\right)\cos\delta_r\right]$$
$$- \left(h_O + h_g\right)\left[\left(F_{y_{f,l}} + F_{y_{f,r}}\right)\sin\delta_f + \left(F_{y_{r,l}} + F_{y_{r,r}}\right)\sin\delta_r\right] \tag{39}$$

$$\delta_{G_z} = l_f\left[\left(F_{y_{f,l}} + F_{y_{f,r}}\right)\cos\delta_f + \left(F_{x_{f,l}} + F_{x_{f,r}}\right)\sin\delta_f\right] - l_r\left[\left(F_{y_{r,l}} + F_{y_{r,r}}\right)\cos\delta_r + \left(F_{x_{r,l}} + F_{x_{r,r}}\right)\sin\delta_r\right]$$
$$+ \frac{t_f}{2}\left[\left(F_{x_{f,l}} - F_{x_{f,r}}\right)\cos\delta_f - \left(F_{y_{f,l}} - F_{y_{f,r}}\right)\sin\delta_f\right] + \frac{t_r}{2}\left[\left(F_{x_{r,l}} - F_{x_{r,r}}\right)\cos\delta_r - \left(F_{y_{r,l}} - F_{y_{r,r}}\right)\sin\delta_r\right] + \sum M_z \tag{40}$$

With $\sum M_z$ is the influence of the self-aligning moments of the tires.

Regarding roll and pitch dynamics, the vehicle's body should be again isolated. This enables the introduction of the suspension forces:

$$\ddot{\phi}\left(I_{x_s} + I_{xz_s}\theta\right) + \ddot{\theta}\phi\left(I_{x_s}\theta - I_{xz_s}\right) - \ddot{\psi}I_{xz_s} = M_s g h_s \phi + \frac{t_f}{2}\left(F_{s_{f,l}} - F_{s2}\right)$$
$$+ \frac{t_r}{2}\left(F_{s_{r,l}} - F_{s4}\right) - (h_O + h_s)\left[\left(F_{y_{f,l}} + F_{y_{f,r}}\right)\cos\delta_f + \left(F_{y_{r,l}} + F_{y_{r,r}}\right)\cos\delta_r\right] \quad (41)$$
$$- (h_O + h_s)\left[\left(F_{x_{f,l}} + F_{x_{f,r}}\right)\sin\delta_f + \left(F_{x_{r,l}} + F_{x_{r,r}}\right)\sin\delta_r\right]$$

$$\ddot{\theta}I_{y_s} = M_s g h_s \theta + l_f\left(F_{s_{f,l}} + F_{s2}\right) - l_r\left(F_{s_{r,l}} + F_{s4}\right)$$
$$+ (h_O + h_s)\left[\left(F_{x_{f,l}} + F_{x_{f,r}}\right)\cos\delta_f + \left(F_{x_{r,l}} + F_{x_{r,r}}\right)\cos\delta_r\right] \quad (42)$$
$$- (h_O + h_s)\left[\left(F_{y_{f,l}} + F_{y_{f,r}}\right)\sin\delta_f + \left(F_{y_{r,l}} + F_{y_{r,r}}\right)\sin\delta_r\right]$$

### 2.4. Model Simplification and Validation

The vehicle equations of motion developed are quite heavy. Our aim is to develop an overall vehicle model so we can synthesize a control strategy. This model should be simple enough but not too simple. To validate the vehicle model, we use as a reference a high-fidelity vehicle model provided by LMS Imagine.Lab AMESim® in a black-box. Complex axle kinematics are used to link the sprung mass and the unsprung one. The procedure is simple. We simulate the high fidelity vehicle model of AMESim® in several scenarios and we compare it with the combination of Equations (12)–(19) and (34)–(42). We identify the order of magnitude of each term in every equation before summing all the components of the equations presented. Then we just simplify the least influencing terms. For a simulation that covers the excitation of all vehicle dynamics, we selected a 3D road reproduced by AMESim's engineers from a real life race track: the approved international Magny–Cours Circuit (https://www.circuitmagnycours.com/). The trajectory tracked at high velocities is illustrated in Figure 5.

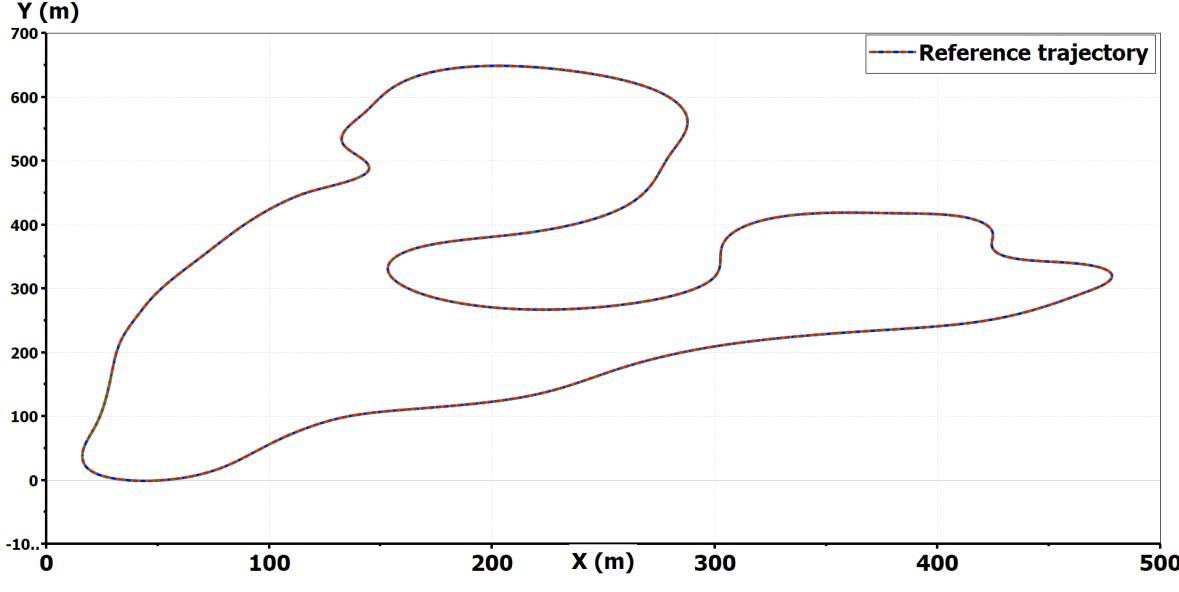

**Figure 5.** Magny–Cours trajectory.

After a careful simplification procedure by simulation, we obtained the following matrix representation:

$$
\begin{bmatrix} \dot{V}_x \\ \dot{V}_y \\ V_z \\ \dot{V}_z \\ \dot{\phi} \\ \ddot{\phi} \\ \dot{\theta} \\ \ddot{\theta} \\ \ddot{\psi} \end{bmatrix} =
\begin{bmatrix}
0 & 0 & 0 & 0 & 0 & 0 & \dfrac{M_s}{M}g & 0 & V_y \\
0 & 0 & 0 & 0 & 0 & 0 & 0 & 0 & -V_x \\
0 & 0 & 0 & 1 & 0 & 0 & 0 & 0 & 0 \\
0 & 0 & 0 & 0 & 0 & 0 & 0 & 0 & 0 \\
0 & 0 & 0 & 0 & 0 & 1 & 0 & 0 & 0 \\
0 & 0 & 0 & 0 & -\dfrac{K_r}{I_{x_s}} & -\dfrac{C_{s_r}}{I_{x_s}} & 0 & 0 & 0 \\
0 & 0 & 0 & 0 & 0 & 0 & 0 & 1 & 0 \\
0 & 0 & 0 & 0 & 0 & 0 & -\dfrac{MK_p + M_s^2 h_s g}{MI_{y_s}} & -\dfrac{C_{s_p}}{I_{y_s}} & 0 \\
0 & 0 & 0 & 0 & 0 & 0 & -\dfrac{M_s^2 h_s g}{MI_z}\phi & 0 & 0
\end{bmatrix}
\begin{bmatrix} V_x \\ V_y \\ z \\ V_z \\ \phi \\ \dot{\phi} \\ \theta \\ \dot{\theta} \\ \dot{\psi} \end{bmatrix}
$$

$$
+
\begin{bmatrix}
\dfrac{1}{M} & 0 & 0 & 0 & 0 & 0 \\
0 & \dfrac{1}{M} & 0 & 0 & 0 & 0 \\
0 & 0 & 0 & 0 & 0 & 0 \\
0 & 0 & 0 & \dfrac{1}{I_{x_s}} & 0 & 0 \\
0 & 0 & 0 & 0 & 0 & 0 \\
-\dfrac{M_s h_s}{MI_{y_s}} & 0 & 0 & 0 & \dfrac{1}{I_{y_s}} & 0 \\
-\dfrac{M_s h_s}{MI_z}\phi & 0 & 0 & 0 & 0 & \dfrac{1}{I_z}
\end{bmatrix}
\begin{bmatrix} F_{x_{tot}} \\ F_{y_{tot}} \\ F_{z_{tot}} \\ M_{x_{tot}} \\ M_{y_{tot}} \\ M_{z_{tot}} \end{bmatrix}
\tag{43}
$$

where:

- z : vertical travel of the sprung mass,
- $V_z$ : vertical velocity of the sprung mass,
- $K_r$ : equivalent overall antiroll bar stiffness,
- $C_{s_r}$ : equivalent overall roll suspension damping,
- $K_p$ : equivalent overall pitch suspension stiffness,
- $C_{s_p}$ : equivalent overall pitch suspension damping,
- $I_z$ : yaw inertia moment of the overall vehicle with respect to its CoG,
- $F_{i_{tot}}$ : combination of tire forces projected at the axis "$i$",
- $M_{i_{tot}}$: combination of moments generated by tire forces with respect to the axis "$i$".

With:

$$F_{x_{tot}} = \left( F_{x_{f,l}} + F_{x_{f,r}} \right) \cos \delta_f + \left( F_{x_{r,l}} + F_{x_{r,r}} \right) \cos \delta_r$$
$$- \left( F_{y_{f,l}} + F_{y_{f,r}} \right) \sin \delta_f - \left( F_{y_{r,l}} + F_{y_{r,r}} \right) \sin \delta_r \tag{44}$$

$$F_{y_{tot}} = \left( F_{y_{f,l}} + F_{y_{f,r}} \right) \cos \delta_f + \left( F_{y_{r,l}} + F_{y_{r,r}} \right) \cos \delta_r$$
$$+ \left( F_{x_{f,l}} + F_{x_{f,r}} \right) \sin \delta_f + \left( F_{x_{r,l}} + F_{x_{r,r}} \right) \sin \delta_r \tag{45}$$

$$F_{z_{tot}} = F_{z_{f,l}} + F_{z_{f,r}} + F_{z_{r,l}} + F_{z_{r,r}} \tag{46}$$

$$M_{x_{tot}} = \frac{t_f}{2} \left( F_{z_{f,l}} - F_{z_{f,r}} \right) + \frac{t_r}{2} \left( F_{z_{r,l}} - F_{z_{r,r}} \right) \tag{47}$$

$$M_{y_{tot}} = l_f \left( F_{z_{f,l}} + F_{z_{f,r}} \right) - l_r \left( F_{z_{r,l}} + F_{z_{r,r}} \right) \tag{48}$$

$$M_{z_{tot}} = l_f \left[ \left( F_{y_{f,l}} + F_{y_{f,r}} \right) \cos \delta_f + \left( F_{x_{f,l}} + F_{x_{f,r}} \right) \sin \delta_f \right]$$
$$- l_r \left[ \left( F_{y_{r,l}} + F_{y_{r,r}} \right) \cos \delta_r + \left( F_{x_{r,l}} + F_{x_{r,r}} \right) \sin \delta_r \right]$$
$$+ \frac{t_f}{2} \left[ \left( F_{x_{f,l}} - F_{x_{f,r}} \right) \cos \delta_f - \left( F_{y_{f,l}} - F_{y_{f,r}} \right) \sin \delta_f \right]$$
$$+ \frac{t_r}{2} \left[ \left( F_{x_{r,l}} - F_{x_{r,r}} \right) \cos \delta_r - \left( F_{y_{r,l}} - F_{y_{r,r}} \right) \sin \delta_r \right] + \sum M_z \tag{49}$$

And:

$$K_r = K_{\phi_f} + K_{\phi_r} \tag{50}$$

$$C_{s_r} = 2c_{s_f} \left( \frac{t_f}{2} \right)^2 + 2c_{s_r} \left( \frac{t_r}{2} \right)^2 \tag{51}$$

$$K_p = 2k_{s_f} l_f^2 + 2k_{s_r} l_r^2 \tag{52}$$

$$C_{s_p} = 2c_{s_f} l_f^2 + 2c_{s_r} l_r^2 \tag{53}$$

where $k_{s_f}$, $k_{s_r}$ are the front and rear suspension stiffness respectively (The front suspensions are alike by design due to the presence of the front steering wheel system. Same remark holds for the rear suspensions.), and $c_{s_f}$, $c_{s_r}$ are the front and rear suspension damping respectively.

Regarding the validation procedure, we make use of a driver model provided by LMS Imagine.Lab AMESim® and designed using a Model Predictive Control (MPC) algorithm to track the Magny–Cours path with an adapted velocity profile. Simulations for this severe maneuver are shown in Figures 6–11.

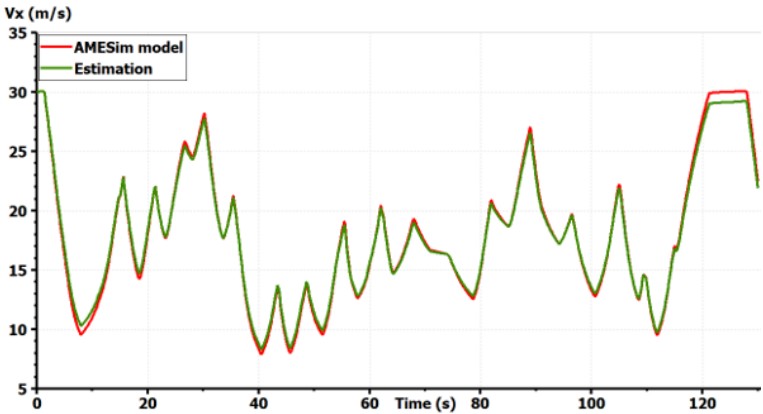

**Figure 6.** Validation of the vehicle model: longitudinal speed [10].

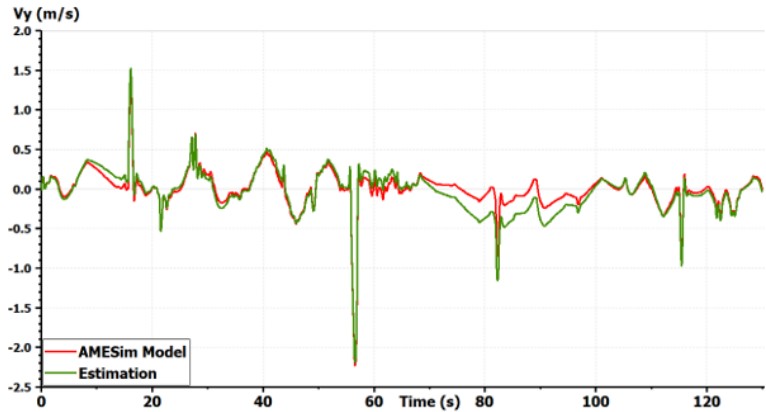

**Figure 7.** Validation of the vehicle model: lateral speed [10].

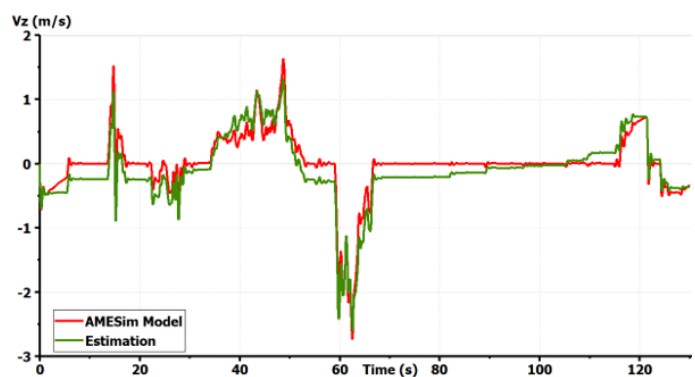

**Figure 8.** Validation of the vehicle model: vertical velocity of the sprung mass [10].

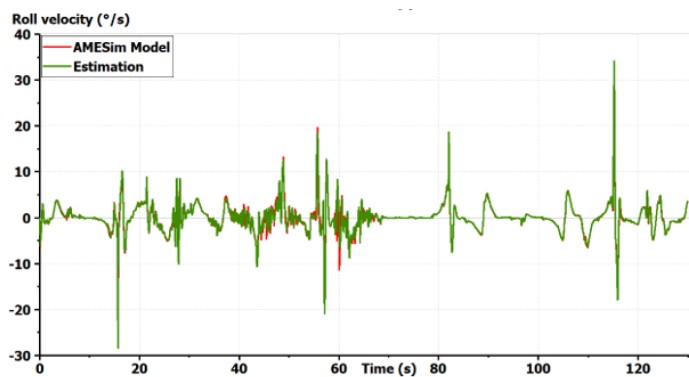

**Figure 9.** Validation of the vehicle model: roll velocity [10].

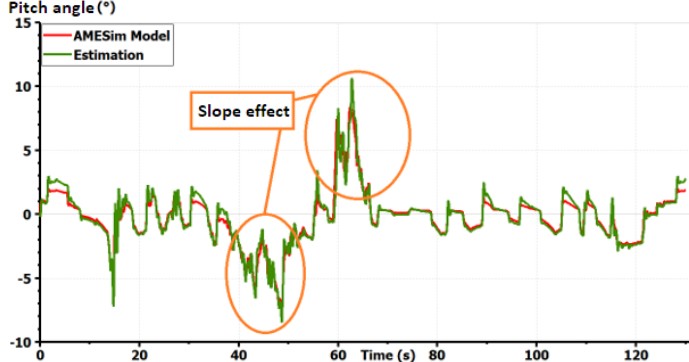

**Figure 10.** Validation of the vehicle model: pitch angle taking into account the slopes [10].

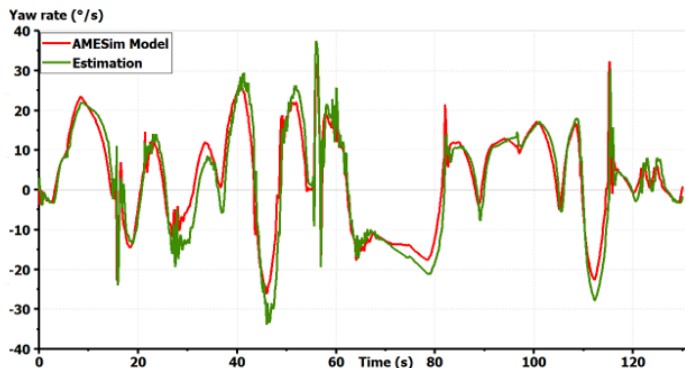

**Figure 11.** Validation of the vehicle model: yaw rate [10].

The model shows good precision for all states in a coupled maneuver. The effect of slopes was also taken into account. This model can then be chosen as a starting model for all GCC synthesis. It is important to start with a complex full vehicle model and then reduce it while justifying each simplification. Starting with a simplified model, as the bicycle model, could lead to the ignorance of important dynamics and couplings making the control fail. If only the horizontal motion is concerned, vertical dynamics can be simplified in the control synthesis. Only couplings between the remaining states should be studied to justify the shape of the controller. However, the vertical forces applied to the tires should always be taken into account as they modify the potential of each tire to drive, brake or steer the vehicle [1].

## 3. Vehicle Motion Control Synthesis

In order to replace the driver and make cars autonomous, additional embedded systems should be implemented. Also, for a car brand to be distinguished from the competition, different subsystems could be implemented by the different manufacturers. One thing is almost certain, future vehicles would be highly over-actuated. In this context, integrated vehicle dynamics control architectures have been reviewed in [13]. The multi-layered architecture appears to be a good candidate to face the future industrial challenges of automotive control. This architecture enables particularly the following criteria: *Adaptability*, *Fault-tolerance*, *Dynamic reconfiguration*, *Extensibility* and *Modularity*.

Here, the vehicle is equipped with an Active Rear Steering (ARS) system, a braking-based Vehicle Dynamics Control (VDC) system, and two rear in-wheel electric motors for Rear Torque Vectoring (RTV). The multi-layered architecture described in Figure 12 is then chosen. The command starts from a lateral behavior target. Here, we simply apply the static expression of the bicycle model as an ideal behavior to aim for [14]. This behavior can be further tuned as in [15] to generate different motion behaviors. The motion of the vehicle's CoG can be ensured by a high-level robust controller. The generalized efforts required to move the vehicle can be distributed in an optimal manner via optimization-based Control Allocation (CA) strategies to the four tires. These tire forces can be then transformed into actuators commands and activate the system concerned avoiding any internal conflicts.

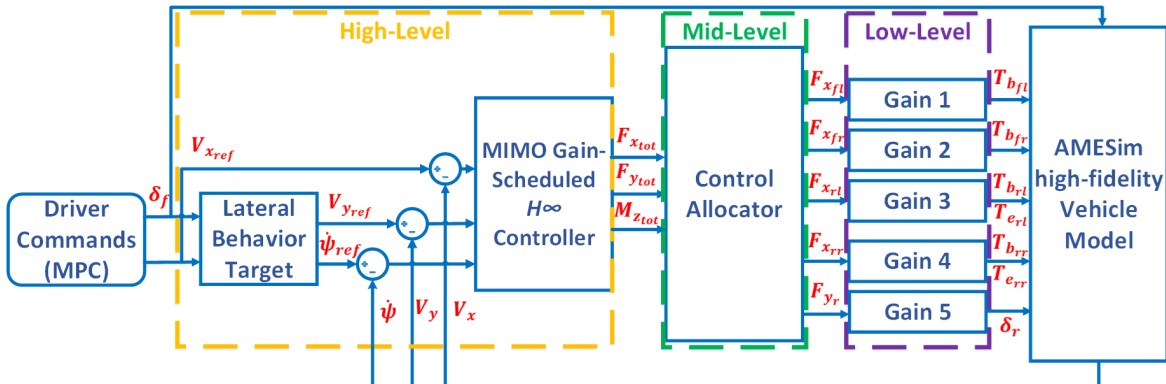

**Figure 12.** Vehicle motion multi-layered control architecture.

### 3.1. High-Level Control

The objective here is to calculate the required forces at the vehicle's CoG in order to track the desired velocities. The dynamics at this point are characterized by inertial parameters as the mass and moment of inertia. These parameters are usually uncertain [5] which require a certain degree of robustness. Moreover, vehicle motion states are coupled as Section 2 demonstrates. A Multi-Inputs Multi-Outputs (MIMO) controller is needed to take into account the different couplings. As the vehicle is equipped by an ARS, VDC, and RTV, and no access is permitted to active suspensions, only a plane vehicle model could be considered at the high-level layer. The importance of vertical dynamics should nevertheless be taken into account at the tire potential estimation [1]. The vehicle high-level state-space representation derived from the global vehicle model exposed in Equation (43) is:

$$
\begin{cases}
\begin{bmatrix} \dot{V}_x \\ \dot{V}_y \\ \ddot{\psi} \end{bmatrix} =
\begin{bmatrix} 0 & 0 & V_y \\ 0 & 0 & -V_x \\ 0 & 0 & 0 \end{bmatrix}
\begin{bmatrix} V_x \\ V_y \\ \dot{\psi} \end{bmatrix} +
\begin{bmatrix} \dfrac{1}{M} & 0 & 0 \\ 0 & \dfrac{1}{M} & 0 \\ 0 & 0 & \dfrac{1}{I_z} \end{bmatrix}
\begin{bmatrix} F_{x_{tot}} \\ F_{y_{tot}} \\ M_{z_{tot}} \end{bmatrix} \\[40pt]
\begin{bmatrix} V_x \\ V_y \\ \dot{\psi} \end{bmatrix} =
\begin{bmatrix} 1 & 0 & 0 \\ 0 & 1 & 0 \\ 0 & 0 & 1 \end{bmatrix}
\begin{bmatrix} V_x \\ V_y \\ \dot{\psi} \end{bmatrix}
\end{cases}
\tag{54}
$$

$$
\tag{55}
$$

As it can be seen, the model is quasi-Linear with Varying Parameters (q-LPV). Effects of the off-diagonal terms should be studied before determining the high-level controller nature. To study the dynamic couplings using classical methods, we first linearize the vehicle model. This is usually done by employing a Taylor series expansion around a nominal system trajectory representing the stable operating points [6]. Using the Jacobian matrix, the model becomes [10]:

$$
\begin{bmatrix} \dot{V}_x \\ \dot{V}_y \\ \ddot{\psi} \end{bmatrix} =
\begin{bmatrix} 0 & \dot{\psi}_e & V_{y_e} \\ -\dot{\psi}_e & 0 & -V_{x_e} \\ 0 & 0 & 0 \end{bmatrix}
\begin{bmatrix} V_x \\ V_y \\ \dot{\psi} \end{bmatrix} +
\begin{bmatrix} \dfrac{1}{M} & 0 & 0 \\ 0 & \dfrac{1}{M} & 0 \\ 0 & 0 & \dfrac{1}{I_{zz}} \end{bmatrix}
\begin{bmatrix} F_{x_{tot}} \\ F_{y_{tot}} \\ M_{z_{tot}} \end{bmatrix}
\tag{56}
$$

where $V_{x_e}$, $V_{y_e}$, and $\dot{\psi}_e$ are the longitudinal velocity, lateral velocity, and yaw rate at the selected stable operating point respectively. Two pre-studies are carried out before moving to the robust control design: the Relative Gain Array (RGA), and Bode diagrams.

### 3.1.1. The RGA

Here, we aim to quantify interactions between inputs and outputs of a MIMO system. The most commonly used technique is the *RGA* developed by Bristol [16]. It helps the controller designer to decide a suitable input/output pairing for the MIMO system. It also gives few hints on pairings to avoid. The RGA of a general non-singular square complex transfer matrix **G** is defined as [10]:

$$RGA(\mathbf{G}\,(i\Omega)) = \mathbf{\Lambda}\,(\mathbf{G}\,(i\Omega)) = \mathbf{G}\,(i\Omega) \circ \left(\mathbf{G}\,(i\Omega)^{-1}\right)^t \tag{57}$$

where "∘" denotes the Hadamard product (Known as elements-by-elements multiplication.), the superscript "*t*" denotes the matrix transpose, and $\Omega$ the considered frequency at which the couplings are studied. Note that the RGA depends on this latter. Therefore, it should be calculated at the crossover frequency chosen by the designer. Discussion about the frequency is provided after the Bode diagrams study. Rules are simple: prefer pairings so that $\Lambda_{ij}$ is close to 1, and avoid pairings with negative $\Lambda_{ij}$. In our case, we use vehicle parameters of a Renault Talisman (Parameters provided by the Group Renault itself.) to first generate a global transfer matrix, and then calculate its RGA. For a crossover frequency of 1 Hz (or $2\pi$ rad/s), we obtain the following matrix:

$$\mathbf{\Lambda}\,(G\,(i2\pi)) = \begin{bmatrix} 1 & 0 & 0 \\ 0 & 1 & 0 \\ 0 & 0 & 1 \end{bmatrix} \tag{58}$$

Which means that the system can be decoupled for a crossover frequency of 1Hz by favoring diagonal pairings. However, for low frequencies, for example $10^{-2}$ rad/s, we find:

$$\mathbf{\Lambda}\,(G\,(i2\pi)) = \begin{bmatrix} 0.05 & 0.95 & 0 \\ 0.95 & 0.05 & 0 \\ 0 & 0 & 1 \end{bmatrix} \tag{59}$$

Which means that off-diagonal terms should be prioritized for both longitudinal and lateral velocities. Another study in the frequency domain is then carried out.

### 3.1.2. Bode Diagrams

To study the importance of frequency for dynamic couplings, we plot bode diagrams corresponding to the linearized model (9). Figures 13–15 show Bode diagrams for the longitudinal velocity, the lateral velocity, and the yaw rate, respectively.

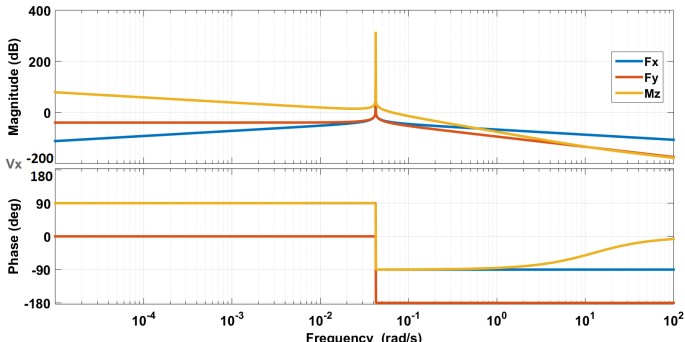

**Figure 13.** Bode diagrams for the longitudinal velocity [10].

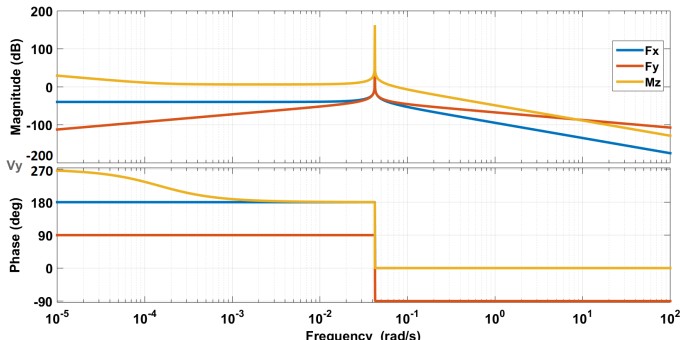

**Figure 14.** Bode diagrams for the lateral velocity [10].

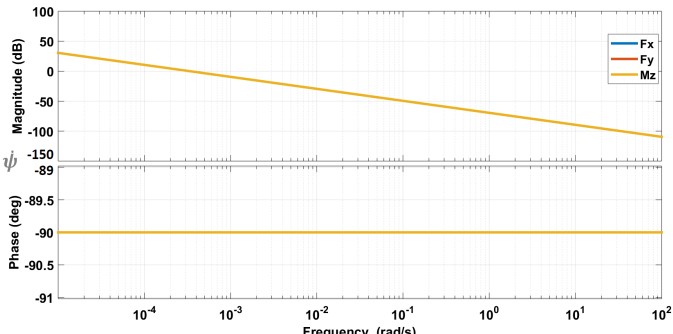

**Figure 15.** Bode diagrams for the yaw rate [10].

Bode diagrams confirm the RGA study. For frequencies higher than 10 rad/s, the influence of $F_{x_{tot}}$ on $V_x$ and $F_{y_{tot}}$ on $V_y$ are preponderant with respect to other inputs. The inverse is observed for low frequencies. The yaw rate can be decoupled for both low and high frequencies. This imposes additional requirement for the high-level controller. In addition of performance and stability, a decoupled controller is preferred for easy tuning. In that case, the crossover frequency should be higher than 10 rad/s.

### 3.1.3. Controller Design

Several robust techniques exist. The Sliding Mode Control (SMC) for example got recently a lot of attention [2,7,8]. This technique however still suffers from several problems as chattering. An optimal design would be preferable. In this context, $\mathcal{H}_\infty$ based design presents several advantages. This technique allows the designer to express explicitly system uncertainties. The biggest advantage of $\mathcal{H}_\infty$ control is that is allows the designer to express explicitly the system's uncertainties. The major idea is to establish a stabilizing controller for a set of uncertain models where the expected model (Called the nominal model.) is contained. The set of uncertain models should not be very large as this may lead to conservatism: a controller able to stabilize a large number of models would have poor performances even for the nominal model. The $\mathcal{H}_\infty$ norm of the plant $G$ is defined as follows:

$$\|G\|_\infty = \sup_{\omega \in [0,+\infty[} |G(j\omega)| \tag{60}$$

This norm can be applied to different specific signals in order to satisfy different requirements. In general, we select an "*augmented plant*" composed of the studied plant itself and weighting functions used to shape the desired transfer functions. Then by applying the $\mathcal{H}_\infty$ norm to the augmented plant, we can enforce the selected signals to follow a desired shape thanks to the weighting functions. A controller, if it exists, is synthesized in this process, and enables to respect all requirements if possible. The problem definition is given below once the objectives are first selected.

$\mathcal{H}_\infty$ drawbacks could be overcome, on one hand, by a different design procedure as it would be shown, and on the other hand, by adding a gain scheduling characteristics.

- **Fixed-structure $\mathcal{H}_\infty$ synthesis**

The main drawback in an $\mathcal{H}_\infty$ control design is the high order of the resulting controller. The order of the controller resulting is equal to the number of states in the plant plus the number of states in the requirements weights plus twice the number of states in the feed through matrix [17]. Here a different methodology is adopted. In the conventional method [18], we first express an augmented plan taking into account tracking errors, control inputs, reference signals, external forces and noises. MIMO performance objectives are then formulated, and weighting functions are defined according to these objectives and added to the augmented system in order to enforce the controller to respect all the objectives. Dynamic or parametric uncertainties can also be added to the augmented plant in order to generate a valid controller to a set of systems and not only the nominal system. This of course enhance the controller robustness, but could lead to the conservatism of the controller performances. A too big augmented plant could lead to a too high-order controller, and too many objectives to fulfill could lead to performance conservatism.

Therefore here, first we by-pass the augmented plant step and we keep only the system (56), and second, we do not express explicitly the uncertainties. Parameters like tire stiffness are highly nonlinear, and it is hard to define a range of variation of such parameters without penalizing the controller performance. This will rather be managed by gain scheduling for vehicle parameters, and adaptive CA for tire parameters using the new Linear with Varying Parameters (LPV) tire model published in [19].

Nevertheless, we add a new requirement to the control design problem, which is the fixed-structure of the controller. An additional effort from the control designer is required. The plant should be studied before choosing between coupled or decoupled control, Proportional-Integral-Derivative (PID) or just PI or phase-lag structure, and so on. This is the goal of the pre-study presented before. According to Figures 13–15, a diagonal controller can be chosen as long as the imposed crossover frequency is higher than 10 rad/s. Moreover, we choose the PI structure for each variable due their integral characteristic at higher frequencies. Six tunable parameters are then chosen in the control design problem. The optimal design algorithm is operated using Matlab®. Both methods can be tested. The conventional method is ensured by the Matlab function "*hinfsyn*" and the fixed-structure method by the function "*hinfstruct*". In this latter, to mitigate the risk of local minima, one could run several optimizations started from randomized initial values of tunable parameters. For more details, see [20].

Regarding performance weighting functions, closed loop shaping is used for defining control design requirements as in [9]. Two objectives are selected: tracking performance, and commands moderation as Figure 16 shows.

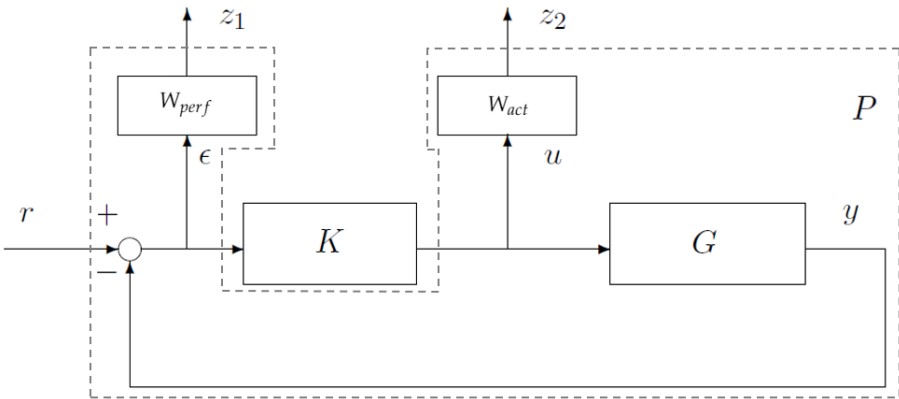

**Figure 16.** Example of a double criteria $\mathcal{H}_\infty$ problem.

For tracking performance, we choose a steady-state offset less than 1%, a closed-loop bandwidth higher than 10 rad/s, and an amplification of high-frequency noise less than a factor 2, which give the weighting function [10]:

$$W_{perf} = \frac{1}{2} \frac{\frac{s}{2\pi 10} + 2}{\frac{s}{2\pi 10} + 0.01} \tag{61}$$

Regarding commands moderation, we use a static gain representing the inverse of the maximum effort, which gives [10]:

$$W_{act} = \frac{1}{1.2Mg} \tag{62}$$

Here we suppose that the maximum friction coefficient is equal to 1.2. The standard $\mathcal{H}_\infty$ problem is then defined as: "*Considering a positive real parameter $\gamma > 0$, find a controller K that satisfies $P \star K$ is asymptotically stable (the real parts of the all the poles of the closed loop system are negative), and $\|P \star K\|_\infty < \gamma$, if it exists, where "$\star$" is the "Redheffer Product".*"

Two well-known design procedures can be adopted: solving a series of Riccati equations, or solving a convex optimization problem under Linear Matrix Inequalities (LMI) constraints. While the first solution is simpler and more reliable but requires the verification of a certain amount of hypothesis, the second one do not rely on these hypothesis but uses a more complex algorithm. The design of the controller is usually done using numerical software as Matlab$^\circledR$. In this latter, both methods can be used (Use the "*hinfsyn*" Matlab function for $\mathcal{H}_\infty$ synthesis. The default method uses Riccati equations. Use 'LMI' as a 'METHOD' in the options field to favour optimization under LMIs constraints.). Details of the algorithms used for the controller synthesis can be found in [18]. The optimization algorithm gives the minimum $\mathcal{H}_\infty$ norm $\gamma = 1.14$ which proves that the different constraints are respected and the high-level controller is stable.

- **Gain-scheduled $\mathcal{H}_\infty$**

One weakness in the preceding design is the dependency on the operating points used to linearize the plant model. For different operating points, different controller parameters values are generated, which influence the controller performance. A proper way to proceed would be to consider different stable operating points and make the controller parameters change with respect to these operating points. This is called scheduling. This consists on considering the nonlinearities in a system as varying parameters. Different linear controllers are designed for each value of the varying parameters. The controller parameters are after automatically adjusted as a function of the varying parameters. In our case, the model (56) is again used. The difference is that $V_{x_e}$, $V_{y_e}$, and $\dot{\psi}_e$ are now considered as varying parameters. As these parameters are also state variables, the system is now q-LPV, which is more challenging [21]. Note that we could have synthesized, from the beginning, a controller using $\mathcal{H}_\infty$ design applied to q-LPV systems. However, as cited in [22], Gain scheduling with respect to direct LPV synthesis techniques presents several practical advantages: better behavior in practice, less conservatism, low computation cost and therefore faster regulations in the design process. As our final objective is to control a real passenger car, we privilege designing separate $\mathcal{H}_\infty$ controllers for the different operating points, and then ensure gain scheduling by simple linear interpolation [23].

The same performances cited in the previous paragraph are pursued for each set of scheduling parameters. The same $\mathcal{H}_\infty$ solver is used for controller robustness. Three-dimensional look-up tables are then generated for each controller parameter. A gain-scheduled $\mathcal{H}_\infty$ controller is then used as in [9] for example. The main difference in the control procedure is the chosen scheduling variables. In [9], a stability index is used to coordinate the subsystems. Here, scheduling is used for dynamic couplings management. Coordination is ensured by optimization-based CA techniques.

*3.2. Mid-Level Control*

The three chassis systems, namely the ARS, VDC, and RTV, can influence the yaw rate, but differently. The VDC can decelerate the vehicle as it is brake-based, while the RTV can accelerate the vehicle and so on. This middle layer aims to coordinate chassis systems in order to avoid conflicts and ensure the generation of the total forces calculated at the high-level layer. To do so, the total forces should be optimally distributed into the four tires to be able to activate the right system with the right amount of effort. As the number of tire forces exceeds the number of the total forces to be applied at the vehicle's CoG, the system is over-actuated, which leads us to a CA problem.

CA problems are defined as follows [24]: *Find the control vector, $\vec{u} \in \mathbb{R}^n$ such that*

$$\mathbf{B}\vec{u} = \vec{d}_{des} \tag{63}$$

*subject to*

$$\vec{u}_{min} \leq \vec{u} \leq \vec{u}_{max} \tag{64}$$

$$\dot{\vec{u}} \leq \dot{\vec{u}}_{max} \tag{65}$$

*where $\mathbf{B} \in \mathbb{R}^{m \times n}$ is a control effectiveness matrix, $\vec{u}_{min}$ and $\vec{u}_{max}$ are the lower and upper position limits respectively, $\dot{\vec{u}}$ is the control rate, $\dot{\vec{u}}_{max}$ is the maximum control rate, $\vec{d}_{des}$ are the desired accelerations, n is the number of control effectors, and m is the number of axes to control ($n > m$).*

For discrete implementation, the rate constraint can be considered as a time-varying magnitude constraint at each sampling interval. This gives the following combined constraints [24]:

$$\underline{\vec{u}} \leq \vec{u} \leq \overline{\vec{u}} \tag{66}$$

where:

$$\begin{cases} \overline{\vec{u}} = \min\left(\vec{u}_{max}, \vec{u} + \Delta t \dot{\vec{u}}_{max}\right) & (67) \\ \underline{\vec{u}} = \max\left(\vec{u}_{min}, \vec{u} - \Delta t \dot{\vec{u}}_{max}\right) & (68) \end{cases}$$

In this way, $\overline{\vec{u}} \in \mathbb{R}^n$ and $\underline{\vec{u}} \in \mathbb{R}^n$ are respectively the most restrictive upper and lower control input limits. $\Delta t$ represents the sampling interval.

In the case of ARS-VDC-RTV coordination, we have:

$$\begin{bmatrix} F_{x_{tot}} \\ F_{y_{tot}} \\ M_{z_{tot}} \end{bmatrix} = \begin{bmatrix} \cos\left(\delta_f\right) & \cos\left(\delta_f\right) & \cos\left(\delta_r\right) & \cos\left(\delta_r\right) & -\sin\left(\delta_r\right) \\ \sin\left(\delta_f\right) & \sin\left(\delta_f\right) & \sin\left(\delta_r\right) & \sin\left(\delta_r\right) & \cos\left(\delta_r\right) \\ b_{3,1} & b_{3,2} & b_{3,3} & b_{3,4} & b_{3,5} \end{bmatrix} \begin{bmatrix} F_{x_{f,l}} \\ F_{x_{f,r}} \\ F_{x_{r,l}} \\ F_{x_{r,r}} \\ F_{y_r} \end{bmatrix} \tag{69}$$

- $b_{3,1} = l_f \sin\left(\delta_f\right) - \dfrac{t}{2} \cos\left(\delta_f\right),$
- $b_{3,2} = l_f \sin\left(\delta_f\right) + \dfrac{t}{2} \cos\left(\delta_f\right),$
- $b_{3,3} = -l_r \sin\left(\delta_r\right) - \dfrac{t}{2} \cos\left(\delta_r\right),$
- $b_{3,4} = -l_r \sin\left(\delta_r\right) + \dfrac{t}{2} \cos\left(\delta_r\right),$
- $b_{3,5} = -l_r \cos\left(\delta_r\right).$

The control vector is then:

$$\vec{u} = \begin{bmatrix} F_{x_{f,l}} \\ F_{x_{f,r}} \\ F_{x_{r,l}} \\ F_{x_{r,r}} \\ F_{y_r} \end{bmatrix} \tag{70}$$

As tires are solicited both longitudinally and laterally, the friction ellipse should be taken into account [1]. Assuming that both vertical forces and friction coefficient are known, we have:

$$\vec{u}_{min} = \begin{bmatrix} -\sqrt{\left(\mu_{f,l}F_{z_{f,l}}\right)^2 - F_{y_{f,l}}^2} \\ -\sqrt{\left(\mu_{f,r}F_{z_{f,r}}\right)^2 - F_{y_{f,r}}^2} \\ -\sqrt{\left(\mu_{r,l}F_{z_{r,l}}\right)^2 - F_{y_{r,l}}^2} \\ -\sqrt{\left(\mu_{r,r}F_{z_{r,r}}\right)^2 - F_{y_{r,r}^2}} \\ -\sqrt{\left(\mu_r F_{z_r}\right)^2 - F_{x_r}^2} \end{bmatrix} \quad \text{and} \quad \vec{u}_{max} = \begin{bmatrix} \sqrt{\left(\mu_{f,l}F_{z_{f,l}}\right)^2 - F_{y_{f,l}}^2} \\ \sqrt{\left(\mu_{f,r}F_{z_{f,r}}\right)^2 - F_{y_{f,r}}^2} \\ \sqrt{\left(\mu_{r,l}F_{z_{r,l}}\right)^2 - F_{y_{r,l}}^2} \\ \sqrt{\left(\mu_{r,r}F_{z_{r,r}}\right)^2 - F_{y_{r,r}^2}} \\ \sqrt{\left(\mu_r F_{z_r}\right)^2 - F_{x_r}^2} \end{bmatrix} \tag{71}$$

To solve the problem, we use repeated optimization performed at every control cycle as it is proposed in [25,26]. This method is computationally demanding as the optimization needs to be carried out at every time-step. Two main criteria should be taken into account. First, the algorithm should provide enough potential to solve the CA (and reallocation in case of a failure in one of the subsystems) and converge to the optimum in a finite number of iterations. Secondly, the algorithm should be fast enough to ensure real-time operations. Various techniques have been tested in the literature. Active Set Algorithms (ASA) have shown good results in this context. Here, the optimal control vector is expressed as [27]:

$$\vec{u}_{opt} = arg \min_{\vec{u}_{min} \leq \vec{u} \leq \vec{u}_{max}} \left\| \mathbf{A}\vec{u} - \vec{b} \right\| \tag{72}$$

$\|.\|$ is a norm that depends on the algorithm adopted to perform the minimization. Two different methods based on ASA have been derived. The first one is called the Sequential Least Squares (SLS). This method uses two stage ASA to separate the global problem into two optimization problems. The predefined order of optimization problems determine the priority of each objective. The second method is the Weighted Least Squares (WLS). This method solves the global problem in one stage ASA by means of different weights to determine the importance of each objective. These weights could be changed online to change the priority of each objective. This can be particularly beneficial if the driver wants to tune the behaviour of the car from for example a sportive mode (favouring RTV) to a comfortable mode (favouring ARS) [15]. Other techniques non based on ASA exist as the Interior Point (IP) solver [28] and the Cascading Generalized Inverses (CGI) [29]. One of the most interesting ones is the Fixed-Point Iteration (FPI) [30] because of its rapidity with respect to other optimization techniques. These techniques are compared in [31] through a co-simulation procedure. It appears that the WLS method is a good compromise between flexibility and rapidity. This method is then used to solve the CA problem in our case:

$$\vec{u}_{opt} = arg \min_{\vec{u}_{min} \leq \vec{u} \leq \vec{u}_{max}} \left\| \mathbf{W_u} \left( \vec{u} - \vec{u}_p \right) \right\|^2 + \gamma \left\| \mathbf{W_v} \left( \mathbf{B}\vec{u} - \vec{d}_{des} \right) \right\|^2 \tag{73}$$

where:

- $\vec{u}_p$ : preferred control vector,
- $\mathbf{W_u}$ : non-singular weighting matrix affecting control distribution among the actuators,

- **W$_\mathbf{v}$** : non-singular weighting matrix affecting the prioritization among the virtual control components when $\mathbf{B}\vec{u} = \vec{d}_{des}$ cannot be attained due to the actuator constraints.

### 3.3. Low-Level Control

Once the forces are optimally distributed, tire forces should be transformed into actuators commands before being fed to any embedded system. Here, engine torques, brake torques, and the rear steering angle should be calculated. This layer represent the most inner loop. Consequently, it should be the fastest one. In this work, rather than using additional dynamic controllers that can complicate the overall design, a static tire model has been preferred which is used as an interface between tire forces and actuators commands. The tire model should take into account the combined slip phenomenon, be precise enough in the controllable zone, and invertible. For these reasons, the LPV tire model [19] has been chosen:

$$F_{x_{i,j}} = C_s^* \left( \alpha_{i,j}, \mu_{i,j}, F_{z_{i,j}} \right) \kappa_{i,j} \tag{74}$$

$$F_{y_{i,j}} = C_\alpha^* \left( \kappa_{i,j}, \mu_{i,j}, F_{z_{i,j}} \right) \alpha_{i,j} \tag{75}$$

With $\kappa$ is the longitudinal slip, $\alpha$ is the side-slip, $\mu$ is the friction coefficient, and where:

$$C_s^* \left( \alpha, \mu, F_z \right) = \frac{4\sqrt{C_s^2 \kappa^{*2} + C_\alpha^2 \alpha^2} - (1 - \kappa^*) \mu F_z}{4 \left( C_s^2 \kappa^{*2} + C_\alpha^2 \alpha^2 \right)} \mu F_z C_s \tag{76}$$

$$C_\alpha^* \left( \kappa, \mu, F_z \right) = \frac{4\sqrt{C_s^2 \kappa^2 + C_\alpha^2 \alpha^{*2}} - (1 - \kappa) \mu F_z}{4 \left( C_s^2 \kappa^2 + C_\alpha^2 \alpha^{*2} \right)} \mu F_z C_\alpha \tag{77}$$

With $C_s$ is the longitudinal stiffness, $C_\alpha$ is the cornering stiffness, $\kappa^*$ and $\alpha^*$ are stable operating points that were chosen in way to ensure that $C_s^* = C_s$ and $C_\alpha^* = C_\alpha$ when no combined slip is involved:

$$\kappa^* = \frac{\mu F_z}{8 C_s^2} \left[ \mu F_z + 4C_s + \sqrt{(\mu F_z)^2 + 8\mu F_z C_s} \right] \tag{78}$$

$$\alpha^* = \frac{\mu F_z}{2C_\alpha} \tag{79}$$

This tire model is then based on varying stiffness to depict the couplings between longitudinal and lateral forces.

Regarding the longitudinal forces, the following Algorithm 1 is adopted:

---
**Algorithm 1** Torques calculation

---
Let $T_{d_{i,j_0}}$ $T_{b_{i,j_0}}$ be starting values
　　$T_{d_{i,j_0}} \leftarrow 0$
　　$T_{b_{i,j_0}} \leftarrow 0$
　　**if** $F_{x_{i,j}} > 0$ **then**
　　　　$T_{d_{i,j}} \leftarrow R_{i,j} F_{x_{i,j}}$
　　　　$T_{b_{i,j}} \leftarrow 0$
　　**else**
　　　　$T_{d_{i,j}} \leftarrow 0$
　　　　$T_{b_{i,j}} \leftarrow -R_{i,j} F_{x_{i,j}}$
　　**end if**

---

With $T_{d_{i,j}}$ and $T_{b_{i,j}}$ are the driving and braking torques at the $i - j$ wheel respectively, and $R_{i,j}$ is the wheels' dynamic radius. Equation (74) serves then just to get a more realistic constraint in

Equation (71) for the CA problem and not to calculate the low-level commands. In contrast, the LPV tire model is used for the calculation of the rear angle:

$$\delta_r = \frac{F_{y_r}}{C_\alpha^*} - \left( \frac{I_z}{Ml_r} - l_r \right) \frac{\dot{\psi}}{V_x}$$

(80)

Equation (80) comes from the defition of the side-slip angle [6]:

$$\alpha_r = \delta_r - \frac{V_y - \dot{\psi}l_r}{V_x}$$

(81)

## 4. Co-Simulation Results

To give reliable results regarding online optimization-based CA, control algorithms have been written in Matlab®, while a high-fidelity vehicle model equipped with an ARS, VDC and RTV systems has been developed in AMESim®. Simulink® is used as a bridge to co-simulate Matlab's high performance algorithms and AMESim's high fidelity vehicle model as Figure 17 shows.

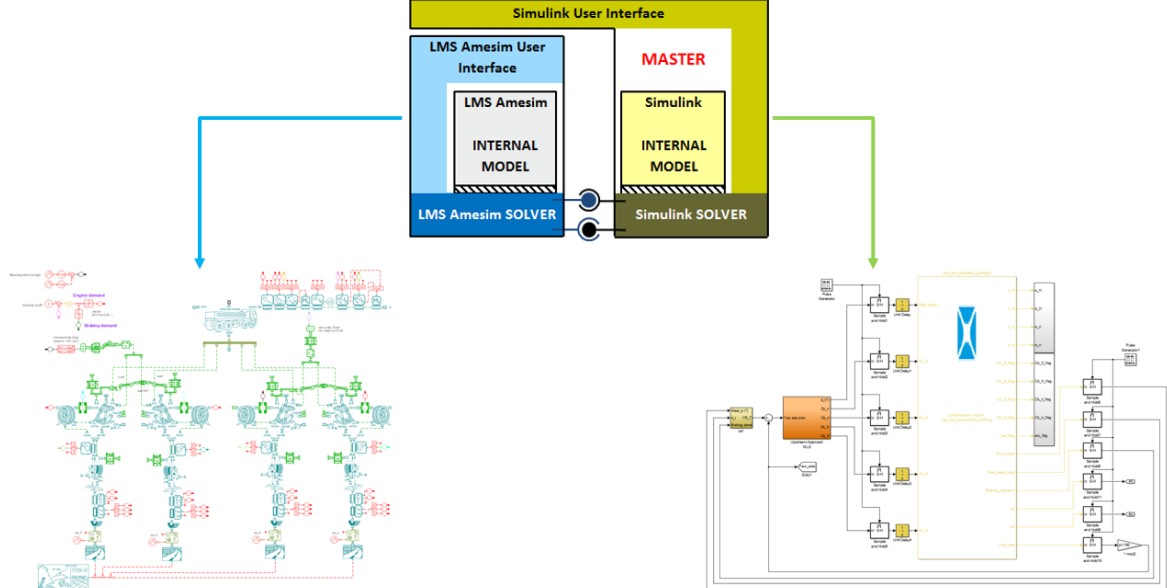

**Figure 17.** Co-simulation procedure.

We will focus mainly on the ISO 3888-1:1999(E) (https://www.iso.org/standard/57253.html) standard, also known as the "VDA test". According to ISO 3888-1:1999(E) (https://www.iso.org/standard/57253.html), the desired speed during all the maneuver should be maintained at 80 km/h. First, we will concentrate on the importance of the crossover frequency. Then the improvement brought by gain scheduling is emphasised. Finally, we show the interest of controlling the lateral velocity.

### 4.1. $\mathcal{H}_\infty$ Controller Only

Here we compare a controller designed at 10 Hz and another one designed at $10^{-1}$ Hz. The longitudinal speed is shown in Figure 18.

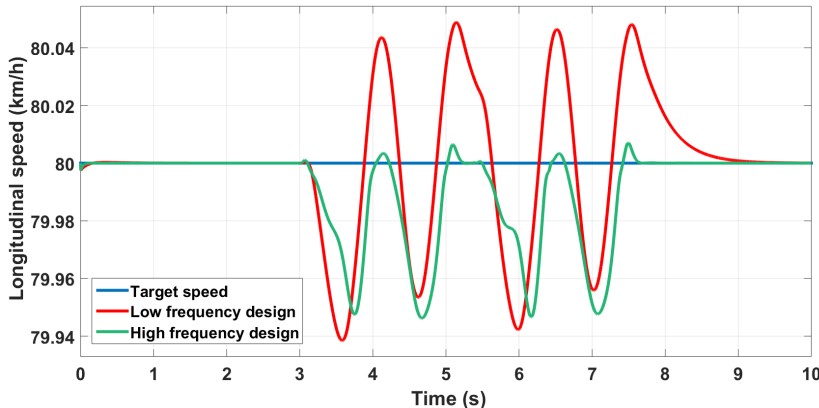

**Figure 18.** Longitudinal velocity control with different crossover frequencies.

The controller designed at a high-frequency is better as expected. A loss of precision and stability is noticed for the low-frequency designed controller especially in the yaw rate control (Figure 19).

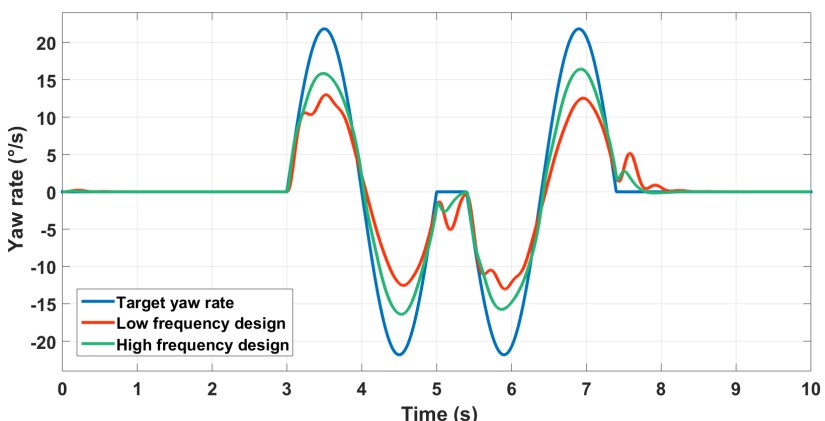

**Figure 19.** Yaw rate control with different crossover frequencies.

To test the robustness of the high-level control, we change the vehicle parameters regarding the mass and inertia by 20%, and wheelbase by 7%, and also tire parameters regarding the cornering stiffness by 40% in AMESim®, while we keep the same parameters of the high-frequency designed controller in Simulink®. We obtain the Figure 20.

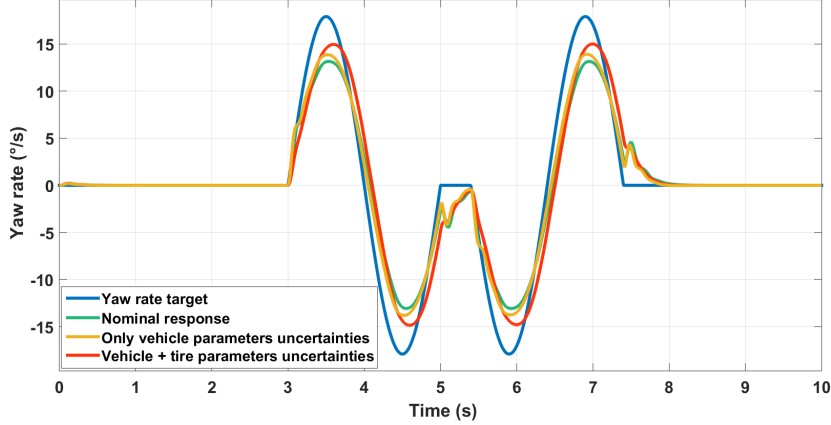

**Figure 20.** Robustness of the controller against parameters uncertainties.

We can see that as long as only vehicle parameters are concerned, the vehicle exhibits almost the same performance regarding yaw rate tracking. When we change considerably the tire cornering stiffness, the uncertainties effects become noticeable. The vehicle behavior remains although acceptable. This was expected as the high-level controller determines the motion of the vehicle's center of gravity. Tire influence is managed rather by the CA and the low-level control. Robustness at these downstream layers should be improved.

We redo the maneuver with the same steering wheel input but with a reduced longitudinal speed of 20 km/h. Results are plotted in Figure 21.

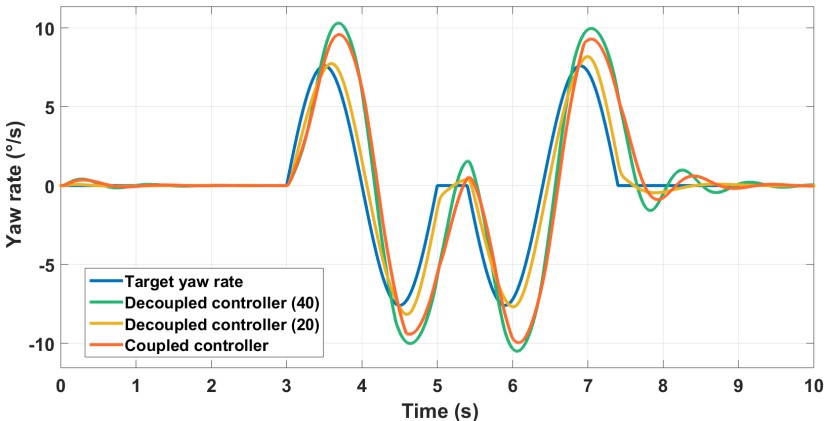

**Figure 21.** Different controllers for the yaw rate.

The controller designed using operating points at a longitudinal speed of 20 km/h exhibits better performance than a controller designed using operating points at a longitudinal speed of 40 km/h for example. This shows that the controller performances are closely related to the operating points used for linearization. Moreover, for these slow dynamics, a coupled controller has the best performance. This confirms the pre-study carried out. Bode diagrams were used to choose the structure of off-diagonal controllers. We can conclude that no fixed operating points can be used for all cases, and no fixed architecture using the $\mathcal{H}_\infty$ only is satisfying.

### 4.2. Gain-Scheduled $\mathcal{H}_\infty$ Controller

Here state-feedback used for closed-loop control and also as scheduling variable for controller parameters. The previous maneuver is repeated for various longitudinal speed values. Satisfying performances regarding the yaw rate control are ensured in different cases where in Figure 22 we show only the performance for two different speed values for more clarity.

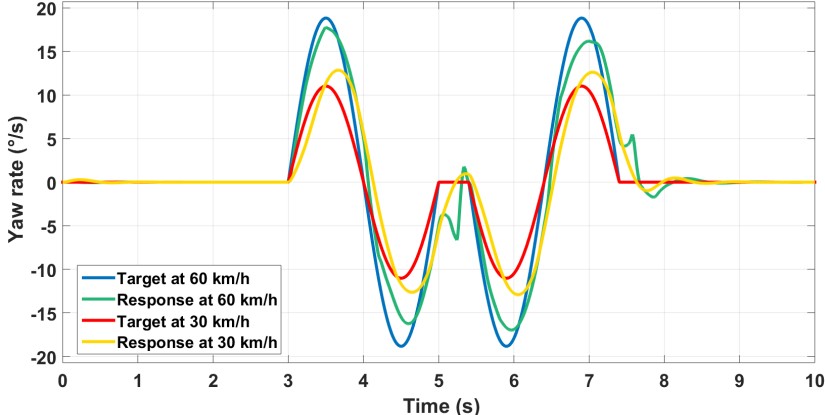

**Figure 22.** Various speed control using GS $\mathcal{H}_\infty$.

However, we can remark the odd behavior of the vehicle at the most difficult dynamics changes, especially when we change rapidly the direction. The behavior also changes from a speed value to another. This may be due to the fact that we only used lookup tables with a basic interpolation algorithm for the different parameters of the controller in the gain-scheduling framework. Another way to tackle the problem is to rather parameterize the controller gains as a polynomial function, and then tune the polynomial coefficients at the different operating points. The order of the polynomial can be increased to add more flexibility. The controller gains may be less accurate at the operating points compared to the lookup tables, but the switch from a behavior to another is softer.

### 4.3. Relevance of Lateral Velocity Control

The Gain-Scheduled controller is used here. Two behaviors are compared. We first use the nominal bicycle model as a reference model for a conventional behavior, and then we minimize the yaw rate while keeping the same target for the lateral speed. The goal is to have a lateral transitional behavior that could be beneficial for obstacle avoidance and stability. The lateral velocity control is illustrated in Figure 23 and the yaw rate in Figure 24.

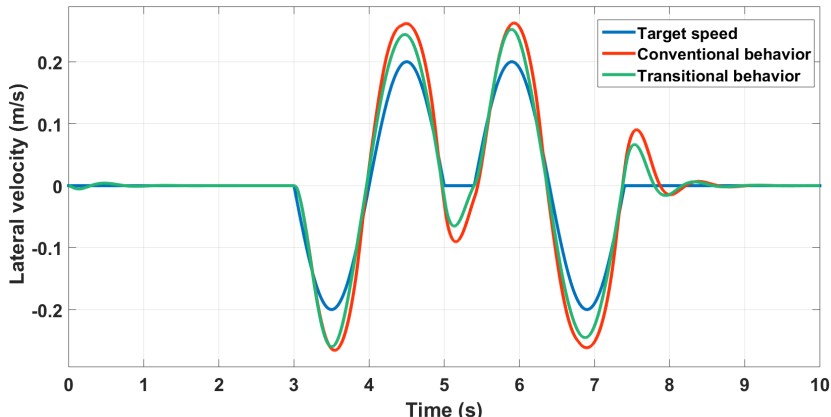

**Figure 23.** Lateral speed control using GS $\mathcal{H}_\infty$.

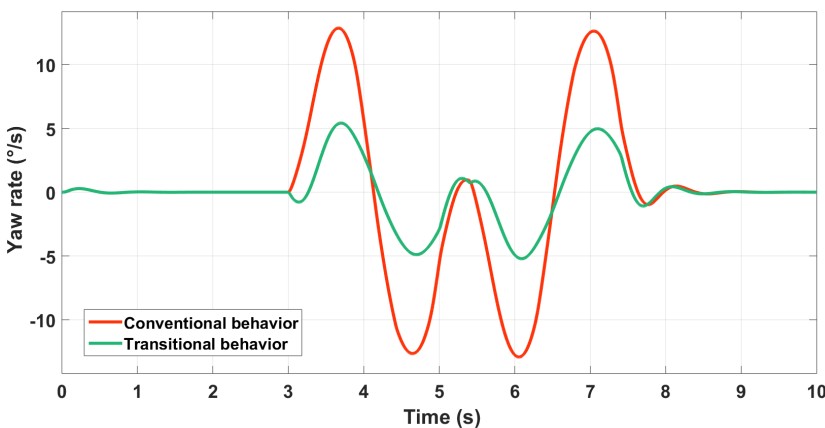

**Figure 24.** Yaw rate control using GS $\mathcal{H}_\infty$.

The controller is able to generate both behaviors by changing the reference. Figure 25 clarifies the difference between both behaviours.

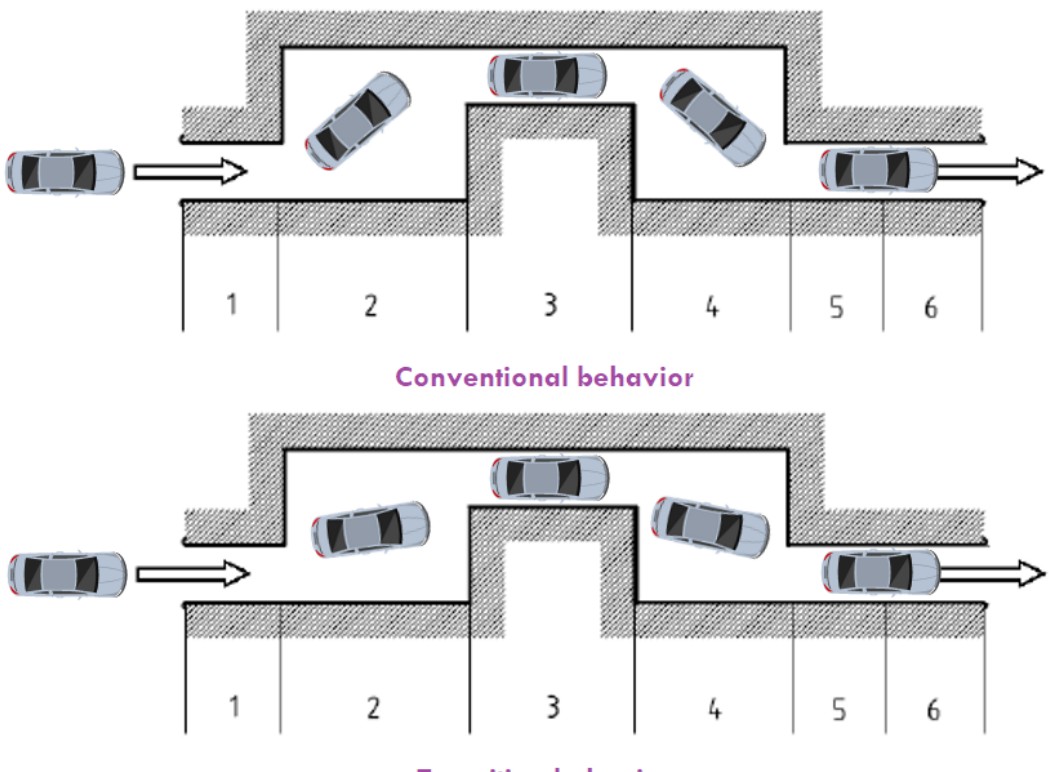

**Figure 25.** Illustration of the difference between motion behaviors.

It should be noted that for a vehicle equipped by rear steering control, stability envelopes should be redefined. In fact, in the literature, stability is more related to the vehicle side-slip angle and its time derivative which is called the $\beta - \dot{\beta}$ phase plane [14]. Here, $\beta = \frac{V_y}{V_x}$ is the vehicle's side-slip angle. A high value of this ratio in conventional vehicles means a loss of control of the vehicle. However, for a four-wheel steering vehicle, this can represent only a lateral transitional behavior.

## 5. Open Challenges

A method of robust adaptive control for over-actuated vehicles has been presented in this paper. Robustness is ensured by means of $\mathcal{H}_\infty$ synthesis, adaptability is established through gain-scheduling, and control distribution is carried using optimization-based CA algorithms. Nonetheless, there are still few challenges to overcome. Particularly, friction changes influence the only effectors that the vehicle has, namely, the tires and induces a nonlinear behavior. In addition, for control distribution, multi-objective problems can be tackled. For autonomous vehicles, qualitative objectives as motion feelings tuning should be satisfied. However, these objectives should be better formalized.

### 5.1. Friction Estimation

One of the major challenges of ground vehicle control is friction estimation. Potential of tires depends on the road/tire interface. This interface is quantified by the coefficient of friction $\mu$ [1]. As it was shown in [32], tire behavior is highly nonlinear and changes with different roads as Figure 26 shows.

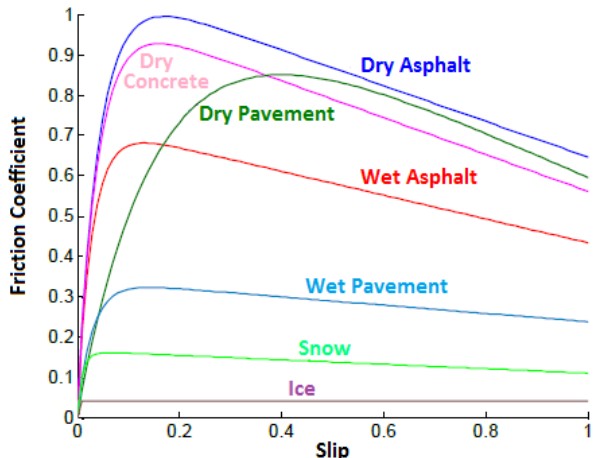

**Figure 26.** Friction coefficient for different surface types [33].

Not only the control should be reconfigurable, but also the reference model should be updated. In [34], a sliding mode observer has been designed depending on the Dugoff tire model [35]. This latter shows good precision in the linear zone of tire force. It does not provide accurate information about the tire/road interface maximum potential. The maximum potential is actually the most needed estimation in vehicle motion control as authors in [36] have cited. In fact, the global chassis control aims to benefit from the overall potential of the vehicle, but without jeopardizing the vehicle stability by exceeding the limits of friction. Recently, data-based techniques have received more attention. For example, an Auxiliary Particle Filter (APF) have been combined with the Iterated Extended Kalman Filter (IEKF) in [37]. Experimental tests showed promising results.

*5.2. Motion Feelings*

Another challenge regarding vehicle motion control is the generated motion feelings. This is extremely important especially for autonomous vehicles, as it would contribute to their acceptance by many people if a confidence feeling is ensured [38] or if in contrast, motion sickness is induced. In this context, over-actuated vehicles present an interesting opportunity to accelerate autonomous vehicles development. In fact, control allocation algorithms offer the possibility to solve online multi-objective problems [27]. One of the objectives could be motion feelings by formulating an acceleration-dependent objective to fulfill [39]. The major problem is that motion has different effects on people with different profiles. The command should be personalized to fit passengers' preferences. This is one of the main reasons of the introduction of the lateral transitional behavior in this paper. It should be noted also that this behavior can be accepted for autonomous vehicles, but can be rather unexpected and, therefore, scary while driving. Indeed, the transitional behavior is an unusual one. Even experienced drivers (According to a first feedback from our partner Renault.) found it surprising as the vehicle seemed to be "sliding", which may lead them to think that they are losing control of the vehicle. Different strategies and behavior references are needed when changing the driving mode.

## 6. Conclusions

In this paper, a global vehicle model has been developed to face GCC challenges. This model is general and can be used for different embedded systems combination. The model has been reduced to cover the case of ARS-VDC-RTV coordination. A gain-scheduled $\mathcal{H}_\infty$ has been synthesized as an adaptive robust high-level controller to specify the vehicle's CoG dynamics. Optimization-based CA algorithms have been used to optimally distribute the control load to the different chassis systems. A co-simulation procedure of Matlab/Simulink$^{\circledR}$ and AMESim$^{\circledR}$ has been carried to obtain more realistic results. The control logic was able to not only to stabilize the over-actuated vehicle, but also to generate new motion behaviors.

This is important for future autonomous vehicles. These vehicles need several embedded systems to operate efficiently, and need, therefore, an optimal inter-systems coordination strategy. In addition, as the driver becomes a passenger, motion sickness should be avoided by generating new motion behaviors to enable comfort and trust feelings.

Our claims still need to be validated through real experiments. A prototype equipped by an ARS, VDC and an RTV is under construction thanks to our partners at the Renault Group. We expect more collaboration from the car manufacturer to expand our research to real-life scenarios. Our future works will probably focus mainly on the implementation of this control strategy into a real vehicle.

**Author Contributions:** Conceptualization, M.K.; methodology, M.K. and D.M.; software, M.K.; validation, M.K., B.M. and X.M.; formal analysis, M.K.; investigation, M.K.; resources, B.M., X.M. and D.M.; data curation, M.K.; writing—original draft preparation, M.K.; writing—review and editing, B.M. and A.T.; visualization, M.K.; supervision, B.M., X.M. and A.T.; project administration, B.M. and X.M.; funding acquisition, X.M.

**Funding:** This research received no external funding.

**Conflicts of Interest:** The authors declare no conflict of interest.

## Abbreviations

The following abbreviations are used in this manuscript:

| | |
|---|---|
| 2WS | 2-Wheel Steering |
| 4WS | four-Wheel Steering |
| AFS | Active Front Steering |
| ARS | Active Rear Steering |
| ASA | Active Set Algorithm |
| CA | Control Allocation |
| CoG | Center of Gravity |
| DoF | Degrees of Freedom |
| ESP | Electronic Stability Program |
| GCC | Global Chassis Control |
| LPV | Linear with Varying Parameters |
| MIMO | Multi-Inputs Multi-Outputs |
| q-LPV | quasi-Linear with Varying Parameters |
| RGA | Relative Gain Array |
| SMC | Sliding Mode Control |
| VDC | Vehicle Dynamics Control |
| WLS | Weighted Least Squares |

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
