# Peer review of "Adaptive Robust Vehicle Motion Control for Future Over-Actuated Vehicles"

_machines, doi:10.3390/machines7020026_

Reviewer 1 Report

This paper is concerned with adaptive robust vehicle motion control for  future over-actuated Vehicles. The paper is hard to follow due to the fact that the authors refer to several definitions and Theorems that are not presented in the paper. The authors should try to write a self-contained paper, as much as possible. Furthermore, it is not clear at this point if Section~2 is a contribution of the paper or not.

The authors claim that the paper is an extended version of a conference paper published in 2018. However, the conference paper is not even cited throughout the text. Moreover, there is no clarification about what is new in the journal version.

There are a lot of papers in the literature concerned with the H-infinity control design problem for linear parameter varying systems. If there is a linear state space representation of the model, I do not understand why the time-varying parameters cannot be considered in the design conditions. There are strategies that can take into account the rates of variation in the time-varying parameters. The authors need to perform a literature review on the H-infinity control problem for LPV and qLPV systems.

It seems that there is no contribution in the control design problem. The authors should write an algorithm to make clear the steps that must be followed to achieve the main goal of the paper.

The authors also could provide supplemental material, including the m.files (with the final model) that have been used to simulate the tests in the paper. The reproducibility of the results at this point is compromised. Some of the controllers obtained should be given in the paper.

To test the robustness the parameters values have been changed. However, it would be more interesting to provide simulations when the parameters are varying with time, since the model studied is a quasi-LPV model as stated by the authors.

There is a lack of comparison with other methods from the literature. It would be interesting to provide a Figure with the control effort.

Minor comments:

There are too many acronyms.

The quality of Figure~3 needs to be improved.

page 3 line 92 "to the the"

The notation in Equation (3) is not clear. The same for (7), (11)... 

The vectors should be represented between brackets.

Figure 11 was not fully explored.

page 15 "Jacobean matrix"

Author Response

Thank you for your precise feedback.

Point 1: The paper is hard to follow due to the fact that the authors refer to several definitions and Theorems that are not presented in the paper. The authors should try to write a self-contained paper, as much as possible. Furthermore, it is not clear at this point if Section~2 is a contribution of the paper or not.

Response 1: Regarding the definitions, the definition of the angular moment has been moved before the definition of the dynamic moment in Page 7 line 126 to facilitate the lecture. A definition of the H-infinity norm has been added in page 17, and the problem is defined in page 18. Regarding the theorems, the theorem in page 5 line 102 has been clarified (it is Newton’s second law of motion). The theorem of Huygens-Steiner has been added in Page 8 line 127. In page 9 line 135, we explain what we mean by “the dynamic moment theorem”.

The Section 2 propose a new global vehicle model. It is a contribution of the paper. The phrase “we propose here a general global vehicle modelling” in page 3 line 82 has been replaced by “we develop here a new detailed global vehicle modelling”.

Point 2: The authors claim that the paper is an extended version of a conference paper published in 2018. However, the conference paper is not even cited throughout the text. Moreover, there is no clarification about what is new in the journal version.

Response 2: The clarification of this point is added in the page 2 line 57. Several citation of the conference paper have been added in corresponding places.

Point 3: There are a lot of papers in the literature concerned with the H-infinity control design problem for linear parameter varying systems. If there is a linear state space representation of the model, I do not understand why the time-varying parameters cannot be considered in the design conditions. There are strategies that can take into account the rates of variation in the time-varying parameters. The authors need to perform a literature review on the H-infinity control problem for LPV and qLPV systems.

Response 3: The goal of our work is to implement practical controllers in real passenger cars for our collaborator Renault. As most of the controllers used today are PI or PDs, we want to propose a smooth roadmap towards more complicated controllers when it is needed. Hence for now a fixed-structured H-infinity is privileged. A better controller is being experimentally tested. Nevertheless, we want to ensure you that we did a literature review on the H-infinity control problem for qLPV systems. In Renault, a thesis has been published in this context: “Applications of Advanced Control Techniques in the Automotive Field” of Dr. Guillermo Pita-Gil in 2011. However, no industrial application has followed the theoritical findings. We added therefore a short comment on why we didn’t use H-infinity LPV from the beginning in page 19, lines 273-279.

Point 4: It seems that there is no contribution in the control design problem. The authors should write an algorithm to make clear the steps that must be followed to achieve the main goal of the paper. 

Response 4: The multi-layered control architecture is being introduced in the automotive sector as a new process design (and not an algorithm). Automotive engineers use a downstream coordination strategy. That means that they develop a controller for each subsystem and then they add coordination strategies downstream the controllers depending on few standardized use-cases. When a subsystem is taken apart, a simplified vehicle model is considered. The tire constraints are often neglected. We do the inverse, we adopt an upstream coordination strategy. This is the reason of the development of a global vehicle model (for a robust high-level control), a “low-cost” control allocation algorithm (originated form the aeronautical field) taking into account tire constraints (the tire model was developed in a previous paper), and then a low-level control to adapt the tire forces requests into actuators commands.

Point 5: The authors also could provide supplemental material, including the m.files (with the final model) that have been used to simulate the tests in the paper. The reproducibility of the results at this point is compromised. Some of the controllers obtained should be given in the paper.

Response 5: I think that the aim of the paper is to enable the reproducibility of the methods exposed so the readers can adapt them for their problems. The objective is to help each car manufacturer to develop its own control logic and not reproduce the same control logic. We provide a methodology to do so and we expect at the end to come up with a standardization of Integrated Vehicle Dynamics Control Architectures. I want to insist on the fact that we work with a car manufacturer and we cannot unfortunately provide the industrial solutions for the upcoming passenger cars. I hope you can understand.

Point 6: To test the robustness the parameters values have been changed. However, it would be more interesting to provide simulations when the parameters are varying with time, since the model studied is a quasi-LPV model as stated by the authors.

Response 6: The parameters considered are the mass, inertia … These parameters can change for example when the vehicle stops and then another passenger or object is added and not in the middle of the manoeuvre. Tire wear can be experienced without the vehicle stopping, but after a very long period, which is not scalable in our simulations.

Point 7: There is a lack of comparison with other methods from the literature. It would be interesting to provide a Figure with the control effort.

Response 7: We want to focus on the design of the fixed-structured H-infinity controller. The figures contain therefore the comparison of the different design choices. Other methods are tested in other papers.

Point 8: Minor comments:

There are too many acronyms.

The quality of Figure~3 needs to be improved.

Page 3 line 92 "to the the".

The notation in Equation (3) is not clear. The same for (7), (11)...

The vectors should be represented between brackets.

Figure 11 was not fully explored.

Page 15 "Jacobean matrix".

Response 8:

We removed the acronyms that have been used only one time: 4WD, ABS, AV, CGI, EPAS, FPI, IP, MPC, PID, SAS, SISO, SLS.

The quality of Figure 3 has been improved.

 The problem "to the the" in Page 3 line 92 has been corrected.

 Clarifications have been added in page 4 in line 98.

 We just used a convention of the cross product. For the ease of the reader, we transformed the bars into brackets as suggested. 

 The quality of Figure 11 (now Figure 12) has been upgraded.

  “Jacobean” have been corrected.

Reviewer 2 Report

A very interesting paper.

Please cite the source of Figure 2 if it is not from the authors of the paper

There is no representation of where points Guf and Gus should be. Based on what criterion should they be defined?

"as you may expect," please remove all the first person/colloquial terms

Eq. 3 probably refers to the acceleration of the roll centre O, but this is not specified

The quality of Figure 3 must be improved

I am not fully clear on how the reference yaw rate is defined

The challenges of friction estimation and motion sickness are important and relevant. What do you think about the additional challenge of vehicle sideslip angle estimation? In the end, the tire slips can only be estimated without a sufficiently precise measurement/estimate of the sideslip angle. The paper would benefit from some comments on this.

Author Response

Thank you for your positive feedback and your advices to improve the paper.

Point 1: Please cite the source of Figure 2 if it is not from the authors of the paper.

Response 1: It is actually our figure. We modified pictures available on the internet.

Point 2: There is no representation of where points Guf and Gus should be. Based on what criterion should they be defined?

Response 2: Figure 2 has been further modified to show approximatively where the CoGs are located. Regarding the unsprung mass, we are quite sure of the position of the CoGs. However, for the sprung mass, it depends on the number of passengers, their weights … which is one of the technical difficulties we are facing.

Point 3: "as you may expect," please remove all the first person/colloquial terms.

Response 3: Done.

Point 4: Equation 3 probably refers to the acceleration of the roll centre O, but this is not specified. 

Response 4: The specification has been added.

Point 5: The quality of Figure 3 must be improved.

Response 5: Done.

Point 6: I am not fully clear on how the reference yaw rate is defined

Response 6: Few lines have been added in page 15 from line 102 to 105 in order to clarify this point with their corresponding references.

Point 7: The challenges of friction estimation and motion sickness are important and relevant. What do you think about the additional challenge of vehicle sideslip angle estimation? In the end, the tire slips can only be estimated without a sufficiently precise measurement/estimate of the sideslip angle. The paper would benefit from some comments on this.

Response 7: Very interesting remark. Thank you. Actually, we benefit from other works from Renault on observers regarding the estimation of not only the sideslip, but even the vehicle’s speed which is only estimated (and therefore also the longitudinal slip …). We are not sure however if we can disclose these information in an open source journal.

Round  2

Reviewer 1 Report

All the comments raised by the reviewer have been properly addressed.